# REVEAL-IT: REINFORCEMENT LEARNING WITH VISIBILITY OF EVOLVING AGENT POLICY FOR INTERPRETABILITY

## ABSTRACT

Understanding the agent's learning process, particularly the factors that contribute to its success or failure post-training, is crucial for comprehending the rationale behind the agent's decision-making process. Prior methods clarify the learning process by creating a structural causal model (SCM) or visually representing the distribution of value functions. Nevertheless, these approaches have constraints as they exclusively function in 2D-environments or with uncomplex transition dynamics. Understanding the agent's learning process in complex environments or tasks is more challenging. In this paper, we propose REVEAL-IT, a novel framework for explaining the learning process of an agent in complex environments. Initially, we visualize the policy structure and the agent's learning process for various training tasks. By visualizing these findings, we can understand how much a particular training task or stage affects the agent's performance in the test. Then, a GNN-based explainer learns to highlight the most important section of the policy, providing a more clear and robust explanation of the agent's learning process. The experiments demonstrate that explanations derived from this framework can effectively help optimize the training tasks, resulting in improved learning efficiency and final performance.

## 1 INTRODUCTION

Reinforcement learning (RL) involves an agent acquiring the ability to make decisions within an environment to maximize the total reward obtained over a series of attempts. The recent achievements in resolving decision-making challenges prove the effectiveness of this paradigm, e.g., video games (Lillicrap et al., 2016; Mnih et al., 2015), robotic controlling (Kaelbling, 1993). Despite many remarkable achievements in recent decades, applying RL methods in the real world remains challenging. One of the main obstacles is that RL agents lack a fundamental understanding of the world and must, therefore, learn from scratch through numerous trial-and-error interactions. Nevertheless, the challenge of verifying and forecasting the actions of RL agents frequently impedes their use in real-world scenarios.

This difficulty is exacerbated when RL is paired with deep neural networks' generalization and symbolic power. Insufficient comprehension of the agent's functioning impedes the ability to intervene when needed or have confidence in the agent's rational and secure behavior. Explainability in RL refers to the ability to understand and interpret the decisions made by an RL agent. Explanations reflect the knowledge learned by the agent, facilitating in-depth understanding, and they also allow researchers to participate efficiently in the design and continual optimization of an algorithm (Miller, 2017). Recent work (Chen & Liu, 2023; Nielsen et al., 2008) in explaining RL models to gain insights into the agent's decision-making process. Some (Deshpande et al., 2020; Bellemare et al., 2017) propose to understand the agent's action by visualizing the distribution of the value function or state value, which is straightforward but limited by the environments, i.e., the distribution of the learned value function is easy to understand in 2D environments but unfeasible in 3D environments.

A line of recent work in explanation and causal RL (Pearl, 1989; Rubin, 1974; Schölkopf et al., 2021) proposes to understand agent's behavior based on a single moment or a single action. Intuitively speaking, a one-step explanation is acceptable, but is it the best way to understand the agent's

behavior? For example, Pearl (2009) proposes learning a causal model to understand what causes an agent to succeed or fail in a given task. However, establishing a causal model with precise causal assumptions in complex environments is challenging and inefficient (Martens et al., 2006; Sainani, 2018; Cui et al., 2020). On the other hand, this becomes more challenging in a long-horizon task as the agent must do hundreds of steps before accomplishing the task. Hence, we debate **the definition of a good interpretability framework for providing explanations in RL**. Firstly, to answer the question, "Why can an agent succeed or fail in a task?". Indeed, we may deduce the pertinent factors from the agent's training process. Similar to how we may make a general assessment of a person's test performance based on their learning process, an agent also requires certain abilities to accomplish a specific task. The agent's mastery of these abilities during the training process will directly impact its ability to effectively complete the task. Conversely, a proficient explanation can enhance the comprehension of the agent's actions and facilitate the agent's improvement of its performance (Schölkopf et al., 2021; Sontakke et al., 2020). Finally, an effective interpretability framework necessitates providing intuitive and comprehensible explanations.

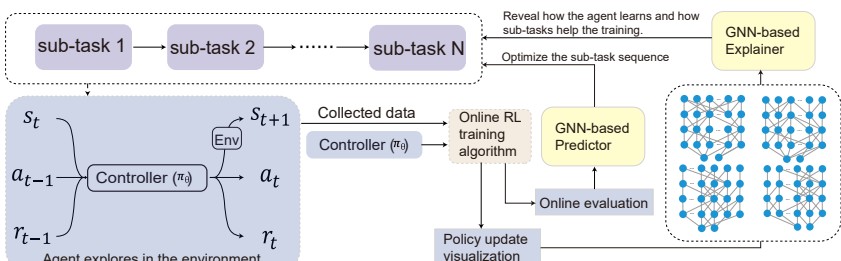

Figure 1: **The main structure of REVEAL-IT.** Assume that we will train an RL agent within an environment to accomplish complex tasks, while directly training the agent on these tasks is challenging and inefficient. In practical application, we will devise sequences of pre-defined sub-tasks (sub-task 1, sub-task 2,..., sub-task N) for training. In REVEAL-IT, we implement the RL agent in the given environment, allowing it to explore and collect data. Subsequently, we train the controller (the control policy $\pi_\theta$), using the collected data. Then, we visualize the policy updates with a node-link diagram. The visualization will depict the structure of the policy and any changes in the weights in the training process. Then, we design a GNN-based explainer to highlight the important changes in the policy updates and a GNN-based predictor to predict the learning progress the agent will achieve in each sub-task. Note that, in this framework, we do not have any limitations on the RL algorithm.

Based on the discussion about a good interpretability mechanism, we propose understanding the agent's performance after training from a more fundamental perspective, i.e., **the learning process of the policy and the sequences of training tasks**. By focusing on the policy and tasks, we may effectively mitigate this problem. They possess a minimum of three evident benefits: **(1)** Both the training tasks and the policy provide well-organized and generalizable information, allowing it to work in diverse environments. **(2)** The information presented in the policy is aligned with the agent's capacities. The limitation of the policy rests in our ability to extract intuitive explanations from the high-dimensional information it provides to understand the actions and behaviors of the agent. **(3)** When compared to SCM or counterfactual methods, which can not deal with the big and complex problems that can not be seen in the real world, using structured training tasks as the goal can solve this problem. This can be further enhanced by extracting relevant information from the policy. At this location, the agent's learning progress can serve as a temporal framework to enhance our comprehension of the agent's behavior.

Considering the preceding discussion, we propose a novel explanation framework, REVEAL-IT. This framework can visually represent updating the policy and the agent's learning process. Additionally, it can offer explanations for the training tasks. This paradigm facilitates a more intuitive comprehension of how agents acquire knowledge in various tasks, identifies which tasks are more efficient in training agents, and explains the reasons for their effectiveness. Simultaneously, this architecture can offer a dependable foundation for developing and optimizing the task sequences. In REVEAL-IT, we first adopt a node-link diagram to visualize the structure of the policy. This design enables humans to directly see the updates made by the RL agent during training. By doing so, we can gain insights into the reasons behind the agent's success or failure during testing. Simultaneously, we can also

perceptibly observe and compare the components of the policy that exist autonomously or are common across several tasks in diverse training scenarios. This offers a visual foundation for aiding the design of training tasks in intricate contexts. Then, We adopt a GNN-based explainer that learns from the policy update process and subsequently analyzes its correlation with the policy's functionality during evaluation. This analysis aims to comprehend the connection between the behaviors performed during training and test. Note that while this explainer will not directly view the data collected from the environment, it is trained in sync with the RL agent. Essentially, we may consider the training of this explainer as a POMDP problem. In the experiments, we test REVEAL-IT in various environments, and the results demonstrate the ability to effectively elucidate the agent's learning process in the training task, optimize the sequence of tasks, and enhance the effectiveness of reinforcement learning.

## 2 RELATED WORKS

**Explainability for RL.** Puiutta & Veith (2020); Heuillet et al. (2020) categorize the explainable RL methods into two groups. The first group is "Post hoc", which aims to provide explanations after the model execution, while the other group is "intrinsic approaches", which inherently possess transparency. Many post hoc explanations, like the saliency map approach (Greydanus et al., 2017; Mott et al., 2019), are based on correlations. This means that conclusions drawn from correlations may not be accurate and do not answer questions about what caused what. However, these algorithms may not be able to model complex behaviors (Puiutta & Veith, 2020) fully. Our method proposes a new mechanism in the problem of explainability for RL from the perspective of the agent's learning process, i.e., it can help to understand how the agent is learning in a given task or environment.

**Explanation in real world.** Deng et al. (2023) provides a general discussion on the challenges posed by reality in real-world scenarios. The non-stationary and constant evolution in the real world requires the RL algorithm to be more robust in handling these changes. Dulac-Arnold et al. (2020); Paduraru et al. (2021) found nine significant problems with RL that make it hard to use in real life. These problems are small sample sizes, unknown or long delays, complex inputs, safety concerns, limited visibility, unclear or multiple reward functions, short delays, learning while offline, and the need to be able to explain. In this paper, we will address the limitations of generating explanations for RL in a high-dimensional environment from the perspective of the training tasks and the learning progress.

**GNN Explainability.** Explainability is a major and necessary characteristic in the field of Graph Neural Network (GNN) study. We primarily focus on GNN explainability strategies that prioritize highlighting the important features of the input rather than offering explanations at the model level. Gradient and feature-based approaches utilize gradients or feature values to evaluate the informativeness of various input features. For example, GRAD (Ying et al., 2019) and ATT (Velickovic et al., 2017) assess the significance of edges by utilizing gradients and attention weights, respectively. Perturbation-based techniques assess the significance of features by analyzing the performance of GNNs when subjected to different input perturbations (Luo et al., 2020; Ying et al., 2019; Yuan et al., 2021). For example, GNNExplainer (Ying et al., 2019) uses a method to learn a mask that explains each prediction individually, whereas PGExplainer (Luo et al., 2020) uses a deep neural network to identify the most critical connections across many predictions. SubgraphX (Yuan et al., 2021) is a perturbation approach that utilizes the Shapley values of features obtained from the MCTS algorithm to identify explanations. One recent method, RGExplainer (Shan et al., 2021), utilizes reinforcement learning to generate an explanation subgraph centered on a chosen beginning point. MATE (Spinelli et al., 2021), a novel meta-explanation technique that aims to enhance the interpretability of GNNs during the training process. The objective is to enhance the comprehensibility of the GNN by iteratively training both the GNN itself and a baseline explanation to acquire an internal representation of the GNN. MixupExplainer (Zhang et al., 2023), similar to our GNN-based explainer, seeks to address the problem of distributional shifts in explanations by combining an explanatory subgraph with a random subgraph. This leads to a clarification that is more closely in line with the graphs observed during the explanation phase.

**Curriculum RL.** Curriculum RL focuses on improving the efficiency and performance of RL agents by organizing training tasks in a meaningful sequence, progressing from simpler to more complex tasks. Previous works (Narvekar et al., 2020; Held et al., 2017) have explored the role of task sequencing in enhancing learning by adapting the difficulty or order of tasks based on the agent's

capabilities. These approaches often rely on heuristics or predefined strategies to design task curricula. In contrast, REVEAL-IT employs a data-driven approach, leveraging a GNN predictor to optimize task sequences based on the agent's real-time learning progress. By aligning task sequences with the agent's evolving policy, REVEAL-IT enhances learning efficiency and provides interpretable insights into how tasks contribute to overall agent performance.

## 3 PRELIMINARIES ON REINFORCEMENT LEARNING AND VISUALIZATION

Our framework comprises of two parts. One is the RL policy visualization module that shows the policy update in training and policy activation in the test. This module is based on fully connected neural networks' well-established node-link diagram representation. The other is a GNN-based explainer, trained to analyze the visualized results in the first module and underly the critical update in the training phase. We can further explore the relationship between different training tasks and the contribution of a specific training task/phase to the final performance. Now, we provide more details on these two parts:

**Reinforcement Learning.** The problem of controlling an agent can be modeled as a Markov decision process (MDP), denoted by the tuple$\{S, A, P, R, \mu_0, \gamma\}$, where $S$ and $A$ represent the set of state space and action space respectively, $P : S \times A \times S \to [0, 1]$ is the transition probability function that yields the probability of transitioning into the next state $s_{t+1}$ after taking an action $a_t$ at state $s_t$, and $R : S \times A \to \mathbb{R}$ is the reward function that assigns the immediate reward for taking an action $a_t$ at state $s_t$. $\mu_0 : S \to [0, 1]$ is the probability distribution that specifies the generation of the initial state, and $\gamma \in [0, 1]$ denotes the discount factor. A is a mapping from states to actions: $\pi : S \to A$. Every episode starts with sampling an initial state $s_0$. At every timestep $t$ the agent produces an action based on the current state: $a_t = \pi(s_t)$, A discounted sum of future rewards is called a return: $R_t = \sum_{i=t}^{\infty} \gamma^{i-t} r_i$. The agent's goal is to maximize its expected return $\mathbb{E}_{s_0}[R_0 \mid s_0]$. In REVEAL-IT, We can use a deep RL algorithm to train the agent (e.g., DQN (Mnih et al., 2015), A2C (Mnih et al., 2016), SAC (Haarnoja et al., 2018b), PPO (Schulman et al., 2017), etc.). In this paper's experiments, we use DQN and PPO as the foundational RL methods. Note that any online RL algorithm can be accepted.

**Strucutral visualization of the policy.** The visualization should meet the following goals: summarize the unique properties of MLP networks in RL and reflect lessons learned from prior tasks. First, the visualization should handle networks of large enough to solve complex tasks. Second, the visualization should depict the network architecture with a node-link diagram. Third, the visualization should allow users to easily experiment with the input-output process of the network, allowing them to judge the network's robustness to translational variance, rotational variance, and ambiguous input. Fourth, the visualization should allow users to view details on individual nodes, such as the activation level, the calculation being performed, the learned parameters (i.e., the node's weights), and the numerical inputs and outputs. We choose to visualize the policy with a node-link diagram, and the only issue with a straightforward implementation of a node-link diagram is that large networks yield a dense mass of edges between layers. We base the visualization model on Harley (2015), which can choose to display some or all of the weights connected to a specific node, and record and correspond to the changes in the value of a specific part of the weights during the training process.

## 4 REVEAL-IT

In REVEAL-IT, assume that we first have a randomly drafted subtask sequence for the RL policy training. We start from the conventional RL training process, i.e., the agent will explore and interact with the environment, trying to complete each sub-task. After the agent collects data, we can apply any online RL algorithm to update the control policy. We can visualize the policy update with a node-link graph $G_O$ by comparing which part of the policy weights are updated. Subsequently, for every fixed number of environmental steps the agent takes, we can compute the true learning progress of the RL agent by evaluating the current policy. Then, we can construct the dataset for GNN explainer training based on the learning progress and graphs of policy updates $G_O$. We formulate the training scheme of REVEAL-IT as in Alg. 1 and introduce more details in REVEAL-IT as follows.

### 4.1 BACKGROUND ON GNN-BASED EXPLAINER

This section establishes the notation and provides an overview of GNN's structured explainer. Assume that we have visualized the learning progress of the RL policy with node-link diagrams following Alg. 1. Let $G_O$ denote the observed diagram on edges $E$ (corresponding to the weights) and nodes $V$ (corresponding to the node in the policy network) associated with $D-$dimensional node features $\mathcal{X} = \{x_1, \cdots, x_n\}$, $x_i \in \mathbb{R}^D$, and $v_i$ represents the $i$-th node in $V$. We use the training data of the RL policy to construct the training set for this GNN explainer. In this set, we will record the update information of the policy $\pi_\theta$, i.e., for a node $v_T^i$ in the policy at time $T$, we denote the weights information connected with this node as $\mathcal{X}_T^i$, and denote the updated weights connected to this node $v_T^i$ as $\mathcal{X}_{T+1}^i$. Thus, we can construct the dataset for training GNN explainer by calculating the updated information by $|\mathcal{X}_{T+1}^i - \mathcal{X}_T^i|$.

The $l$-th layer of the GNN explainer performs three essential computations: MESSAGE, AGGREGATE, and UPDATE. The model first calculates the messages between every pair of nodes by $m_{ij}^l = MESSAGE(h_i^{l-1}, h_j^{l-1}, r_{ij})$, where $h_i^{l-1}$ and $h_j^{l-1}$ denote the node representations of $v_i$ and $v_j$ obtained from the previous layer $l-1$, respectively, and $r_{ij}$ represents the relation between them. Next GNN aggregates the messages for each node $v_i$ by $m_i^l = AGGREGATE(m_{ij}^l|v_j \in \mathcal{N}(v_i))$, where $\mathcal{N}(v_i)$ refers to the neighborhood of $v_i$. Finally, the GNN explainer updates the node representation from the previous layer by $h_i^l = UPDATE(m_i^l, h_i^{l-1})$. The final embedding for node $v_i$ after $L$ layers of computation is $z_i = h_i^L$, which is then used for node classification, i.e., the understanding the critical nodes in the RL agent's evaluation is a crucial pre-requisite for determining the significance of weights updating.

---

**Algorithm 1** REVEAL-IT

1: **Initialize:** control policy $\pi_0$, RL replay buffer $\mathcal{B} \leftarrow \emptyset$, policy updated dataset $\mathcal{D}_p \leftarrow \emptyset$, a set of $N$ training tasks $\mathcal{D}_{task}$, randomly sampled sequence of training tasks $Seq_0$, the predicted learning progress $\{\mathcal{P}(task_n, \pi_0) = 0 | n = 1, 2, \cdots, N\}$;
2: **for** $t$ in interations **do**
3:     $p$ = random();
4:     **if** $p < \epsilon$ **then**                            ▷ Sample training task sequence with $\epsilon-$greedy
5:         Randomly sample training task sequence $Seq_t$ from $\mathcal{D}_{task}$;
6:     **else**
7:         Sample training task sequence $Seq_t$ in terms of $\{\mathcal{P}(task_n, \pi_t)\}$;
8:     **end if**
9:     **for** $task_i$ in $Seq_t$ **do**
10:         RL agent explores and collects MDP data to $\mathcal{B}$ to complete $task_i$;         ▷ RL exploration
11:         Use any online RL algorithm to update $\pi_0$ using samples from $\mathcal{B}$;
12:         Evaluate the current policy $\pi_t$ and compute $\mathcal{P}(task_i, \pi_t)$ following the Eq. 1;
13:         Visualize policy update information with graph $G_{O,t}$;              ▷ Visualization
14:         Save $(G_{O,t}, \mathcal{P}(task_i, \pi_t))$ into $\mathcal{D}_p$;        ▷ Construct dataset for GNN learning
15:     **end for**
16:     Train GNN predictor to minimize $|\hat{\mathcal{P}} - \mathcal{P}|^2$ based on $\mathcal{D}_p$;
17:     Train GNN explainer to partition $G_{O,t}$ into the optimal subgraph $G_{X,t}^m$;
18: **end for**

---

### 4.2 REVEAL-IT

**The learning objective of GNN predictor.** The overall goal of the GNN explainer is to learn to optimize the sequences of training tasks to improve the learning efficiency of the RL agent. We take inspiration from the learning progress in earlier work on curriculum learning (Narvekar et al., 2020; Ao et al., 2023), i.e., a higher learning progress signifies that the agent can acquire more knowledge regarding a specific training task, reflecting a higher learning efficiency. Therefore, in REVEAL-IT, we train the GNN predictor to learn to predict how much improvement the RL agent can have in a given task. Based on the discussion in Sec. 4.1, we denote the prediction task of our GNN explainer as $\Phi: G_O \to \hat{\mathcal{P}}$, where $\hat{\mathcal{P}}$ is the predicted improvement of the RL policy. Thus, in a given task $n$, we train the GNN predictor to minimize the error between $\hat{\mathcal{P}}$ and the true learning progress $\mathcal{P}$ defined as:

$$\mathcal{P}(task_n, \pi_t) = \mathcal{R}(task_n, \pi_t) - \mathcal{R}(task_n, \pi_{t-1}), \tag{1}$$

where $\mathcal{R}(task_n, \pi_t)$ denotes the expected return that the RL agent achieves with policy $\pi_t$.

**Train GNN explainer with RL agent simultaneously.** In REVEAL-IT, we separately train a GNN explainer that helps to understand how an RL agent learns to complete tasks, i.e., the structural visualization of the policy allows us to observe which part of the policy weights is updated in one iteration, and the GNN explainer learns to highlight the most crucial updates related to the agent's final success. One thing to note is that the training data used for the GNN explainer is based on the RL, which allows the GNN explainer to adapt to the learning process of the RL agent and can provide guidance during the training. More specifically, as the RL agent acquires MDP data from the interactions with the environment and learns from it, we can use any online RL algorithm to update the RL policy. After each policy update, we can obtain a structured update visualization of the policy by comparing the changes in the policy weight parameters and constructing the dataset for the GNN explainer based on the updated information. We can regard the visualized policy as a node-link graph, i.e., the updated weights can be viewed as the edges that connect with nodes in the graph. Thus, the learning objective of the GNN explainer can be described as "learn to find which part of the edges (updated weights) is more important for the success of the RL agent". Now, we introduce more details of the GNN explainer to help the understanding.

**Step 1. Understanding the visualized policy.** We first train the GNN explainer on the learning data collected from the RL. During the RL evaluation, the activated nodes in the policy will be tagged and utilized as the ground truth for the GNN explanation.

Note that the active nodes during evaluation will vary as the RL policy training progresses. This variability may raise issues regarding the stability of GNN's training. However, this problem is non-existent. This is because we require the GNN explainer to evaluate the significance of the training job by considering the RL agent's present level of learning progress. The active nodes in the evaluation and the collected environment reward reflect the capabilities of the current policy. When the policy is updated, especially when the agent's ability is increased, the different activated nodes and the increase in the environment reward can give the GNN predictor the information it needs. This enables the predictor to acquire knowledge on predicting the agent's improvement based on the current learning process. Consequently, the GNN predictor can more precisely assess the value of the current training task.

**Step 2. Highlight the important updates for explanations.** As mentioned in Sec. 3, the control policy can be too complex to understand humans, even with the visualized diagrams of the policy. Therefore, we also train the GNN explainer to highlight these important nodes and their related updates to humans to achieve a feasible explanation. The GNN explainer aims to partition $G_O$ into two subgraphs:

$$G_O = G_X + \Delta G, \tag{2}$$

where $G_X$ represents the explanatory subgraph reflecting the important edges (corresponding to policy updates), and $\Delta G$ is the remaining graph containing unimportant edges. When explaining a graph of visualized policy updates, we denote the optimal subgraph by $G_X^m$. When $G_X^m$ is used as input to $\Phi$, it can retain the original prediction. $G_X^m$ is optimal, as removing the features from it would result in a different prediction.

**The learning objective of GNN explainer.** In REVEAL-IT, We use the loss in the PGEexplainer (Luo et al., 2020) to train the GNN explainer, and we use an MSE loss for the GNN predicter. On the other hand, in the traditional GNN explainer method, the learning objectives of the predictor (evaluator) GNN and the explainer are often consistent, i.e., maximizing the mutual information between the label and the selected subgraph, and we can express this process as:

$$\max_{G_s} = MI(Y, G_S) = H(Y) - H(Y|G = G_S), \tag{3}$$

where $G_S$ denotes the masked subgraph, the GNN explainer optimizes this function locally by training an NN to provide a customized explanation for each instance. However, REVEAL-IT optimizes this by adopting an NN to learn the crucial edges on multiple instances (each evaluation result of RL).

**The different roles of GNN explainer and predictor in REVEAL-IT.** The GNN predictor and explainer serve distinct roles in REVEAL-IT. The former assesses the improvement of the training task on the RL policy performance, thereby increasing the learning efficiency, whereas the latter evaluates whether the GNN predictor comprehends the learning process of the RL agent by analyzing the correlation between "nodes linked to significant updates" and "the activated nodes during the

test". Simultaneously, by analyzing the graphs of the policy update by the GNN explainer, we can ascertain the ability learned by the RL agent in a specific task and comprehend the significance of this capability in the RL agent's success in the final task. REVEAL-IT is distinguished from other explainable RL methods by its independence from external data and structure. It learns to explain the learning process of the RL agent within the training task without imposing constraints on the environment or specific RL algorithms.

## 5 EXPERIMENTS

We design our experiments to answer the following questions: (1) Can REVEAL-IT show the learning process of an RL agent in a given environment? (2) Can REVEAL-IT improve the learning efficiency of the agent based on the explanations? Our project can be viewed by: REVEAL-IT.

### 5.1 EXPERIMENTS SETUP

**Environments.** We base our experiments on two types of benchmarks. The first one is **ALFWorld benchmark** (Shridhar et al., 2021), a cross-modality simulation platform encompassing a wide range of embodied household tasks. ALFWorld combines a textual environment with a visual environment that the Ai2Thor simulator (Kolve et al., 2017) renders for each task. This textual part uses the Planning Domain Definition Language (PDDL) (Ghallab et al., 1998) to turn each pixel observation from the simulator into a text-based observation that is equal to it. It then uses the TextWorld engine (Côté et al., 2018) to create an interactive environment. The tasks within the ALFWorld benchmark are categorized into six types: Pick &Place, Clean & Place, Heat & Place, Cool & Place, Look in Light, and Pick Two Objects & Place. Each task requires an agent to execute a series of text-based actions, such as "go to safe 1", "open safe 1", or "heat egg 1 with microwave 1", following a predefined instruction. These actions involve navigating and interacting with the environment. The other benchmark is evaluated in 6 OpenAI RL benchmark domains (Brockman et al., 2016), which are commonly adopted in RL tasks.

**The complex tasks in ALFWorld.** In ALFWorld, accomplishing a given task requires the agent to complete a sequential arrangement of sub-tasks step by step (as shown in Fig. 2). A task may entail interactions with more than 10 items and necessitate over 30 steps for a human expert to solve, thereby testing an agent's ability in long-term planning, following instructions, and using common knowledge. To facilitate a thorough comprehension, we have included an instance of each task category in the appendix in Fig. 4. Using the original tasks directly to train an RL agent is inefficient. Therefore, it is more suitable to use subtask training intuitively. However, this raises the question of how to structure the subtask sequence, specifically if the subtask sequence should remain consistent throughout the training process. What is the criteria for adjusting the subtask sequence? The crucial factor is determining the optimal subtask sequence that aligns with the agent's current training progress.

**Baselines.** The fundamental inquiry of REVEAL-IT is whether it can operate effectively in complex tasks and environments, i.e., does it have the ability to enhance the training efficiency of RL significantly? Can it be modified to suit various reinforcement learning techniques? Hence, we will focus on answering the first question in the ALFWorld. The baseline methods include MiniGPT-4 (Zhu et al., 2023), BLIP-2 (Li et al., 2023), LLaMA-Adapter (Gao et al., 2023) and InstrctBLIP (Dai et al., 2023). They share state-of-the-art and competitive performance in ALFWorld and solely interact with the visual world, aligning with the approach of REVEAL-IT. Our primary emphasis in the OpenAI RL benchmark was on the third question. We use PPO (Schulman et al., 2017), SAC (Haarnoja et al., 2018a), DQN (Mnih et al., 2015) as the fundamental RL algorithms to assess if REVEAL-IT can enhance training efficiency and performance across various approaches.

### 5.2 MAIN RESULTS

**Visualization of the learning process in ALFWorld.** In Figure. 2, we visualize the learning process of the RL policy in ALFworld. The RL policy we used in ALFworld is designed as an actor-critic structure consisting of two multi-layer prediction (MLP) networks as "actor" and "critic". Considering that when we evaluate or test the RL agents, the "actor" is responsible for making decisions, we primarily focus on showcasing the learning process of this aspect of the policy. The actor-network

consists of 4 fully connected layers, each including 64 nodes. We implemented ReLU function on the first and third layers to visually identify the activated nodes throughout the evaluation process.

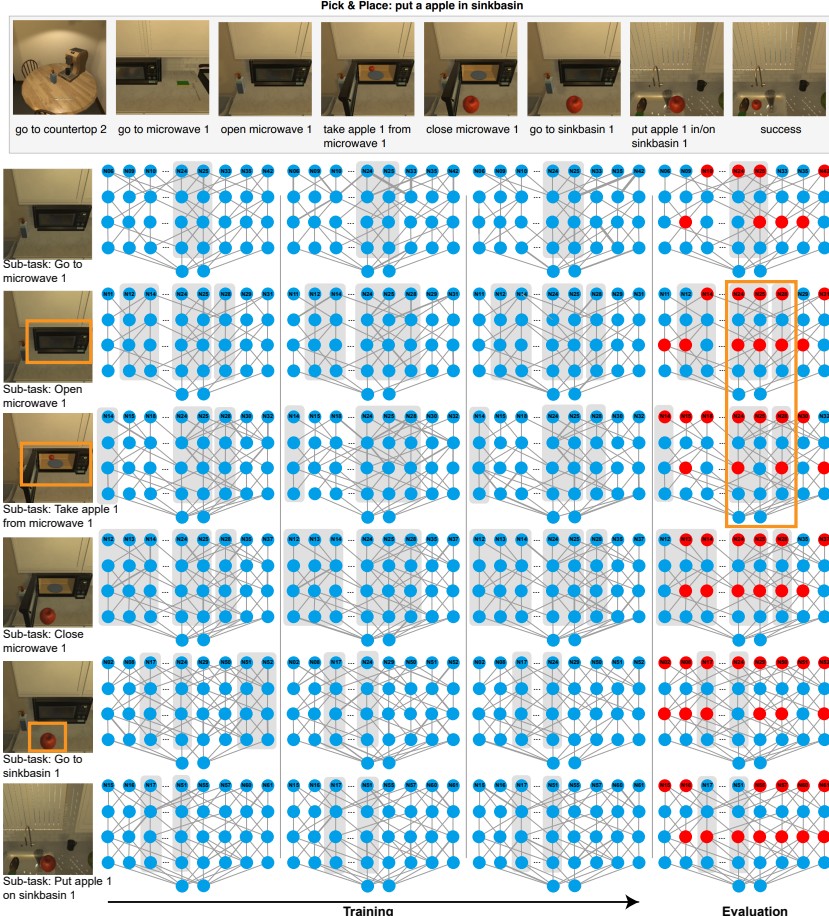

Figure 2: **Visualized important policy updates by GNN explainer.** We provide a larger version in Appendix. E, and detailed analysis in section 5.3. In this figure, the first line depicts the sequential arrangement of sub-tasks that must be accomplished step-by-step to accomplish a given task fully. One of the sub-tasks is located on the left side of the second line. The first to third columns of the tree diagram illustrates the RL policy update process. The blue circles represent nodes in the neural network, while the connections between the circles represent weight updates. Thicker connections indicate larger updates in weight amplitude (selected by GNN explainer). The red circles in the tree diagram on the far right illustrate the specific policy nodes that are active during the evaluation process. The links here represent the revised weights throughout the training phase of this subtask. The orange squares indicate the portions of the policy that are common to several sub-tasks. We opt to depict the 8 interconnected nodes with the most significant weight adjustment, facilitating comprehension of the reinforcement learning policy's learning process and highlighting the policy's shared component more distinctly.

**REVEAL-IT brings improvement.** We report the performance of REVEAL-IT on the ALFWorld benchmark in Tab. 1 and compare it to the performance of 7 other representative agents. We report the performance on the OpenAI benchmark in Tab. 2. Given that REVEAL-IT exclusively engages with the visual engine within ALFWorld, the baseline agents similarly only interact with the visual environment, aligning with their configurations as stated in the original paper. We evaluate the primary metric: the success rate, defined as the proportion of completed trials. REVEAL-IT demonstrates significantly better performance than other VLM agents, establishing its superiority. Notably, REVEAL-IT does not rely on LLM agents, unlike other baseline models that utilize pre-trained LLM agents to generate planning steps using ALFWorld's text engine during the training phase. This demonstrates that the GNN-based explanation efficiently optimizes the sequence of tasks and significantly enhances the agent's training.

| Agent | Success Rate | | | | | | |
|---|---|---|---|---|---|---|---|
| | Avg. | Pick | Clean | Heat | Cool | Look | Pick2 |
| ResNet- 18* (Shridhar et al., 2020) | 0.06 | - | - | - | - | - | - |
| MCNN-FPN* (Shridhar et al., 2020) | 0.05 | - | - | - | - | - | - |
| MiniGPT-4 (Zhu et al., 2023) | 0.16 | 0.04 | 0.00 | 0.19 | 0.17 | 0.67 | 0.06 |
| BLIP-2 (Dai et al., 2023) | 0.04 | 0.00 | 0.06 | 0.04 | 0.11 | 0.06 | 0.00 |
| LLaMA-Adapter (Gao et al., 2023) | 0.13 | 0.17 | 0.10 | 0.27 | 0.22 | 0.00 | 0.00 |
| InstructBLIP (Dai et al., 2023) | 0.22 | 0.50 | 0.26 | 0.23 | 0.06 | 0.17 | 0.00 |
| PPO (Schulman et al., 2017) | 0.04 | 0.01 | 0.05 | 0.04 | 0.07 | 0.03 | 0.00 |
| **REVEAL-IT** (Ours) | 0.80 | **0.66** | **0.90** | **0.81** | **0.80** | **0.85** | **0.70** |
| **Human Performance* (Shridhar et al., 2020)** | 0.91 | - | - | - | - | - | - |

Table 1: **Comparison with the SOTA agents in ALFWorld.** * - reported in the previous work. The highest scores for each task in the same type of environment are highlighted in bold. REVEAL-IT substantially outperforms other SOTA agents of interacting with visual environments, and its success also directs a promising way to achieve human-level performance in ALFWorld.

**The optimization of the training task sequences.** We report the distribution of the sub-tasks in Fig. 3. As the training progresses, it becomes evident that the GNN-based explainer modifies the distribution of sub-tasks. Fig. 3 (left) shows the distribution of randomly sampled subtasks at the start of the training process. It is evident that the subtask "put" occurs most frequently. During the training, while examining the distribution of subtasks in Fig. 3 (middle left), task "put" remains the most common subtask; the frequencies of "look", "pick", and "find" have experienced a substantial increase. This demonstrates that these three qualities are important for the agent in ALFWorld. This aligns with our inherent comprehension of the environment, wherein the agent must initially acquire the ability to locate and retrieve the relevant item before it can proceed with future activities. As the training advances, the occurrence rate of these three task types declines proportionally (Fig. 3 (middle right)), suggesting that the agent already possesses the necessary skills. Training the agent with these tasks will not significantly enhance its performance. Fig. 3 (right) demonstrates that as the training advances, the training tasks increasingly emphasize the agent's additional skills, which align with the original work's final one or two stages. This indicates that the agent's training is almost over, and the alterations in the distribution of the training task align with our comprehension of the agent's learning process. It is evident that the explainer has comprehended and become proficient in the agent's learning process, enabling them to modify the training tasks effectively.

| Agent | Environment | | | | | |
|---|---|---|---|---|---|---|
| | HalfCheetah | Hopper | InvertedPendulum | Reacher | Swimmer | Walker |
| PPO | 1846.25 (1.00) | 2250.46 (1.00) | 986.85 (1.00) | -10.34 (1.00) | 108.81 (1.00) | 2954.01 (1.00) |
| PPO+REVEAL-IT | **1921.08 (0.90)** ↑ | 2104.88 (0.90) | **1004.92 (0.90)** ↑ | -11.27 (0.90) | **112.51 (0.90)** ↑ | **3148.64 (0.90)** ↑ |
| A2C | 1014.02 (1.00) | 630.51 (1.00) | 1002.45 (1.00) | -27.02 (1.00) | 25.28 (1.00) | 676.52 (1.00) |
| A2C+REVEAL-IT | **1147.89 (0.80)** ↑ | **742.17 (0.80)** ↑ | 966.20 (0.80) | -28.54 (0.80) | 17.63 (0.80) | **810.26 (0.80)** ↑ |
| PG | 602.27 (1.00) | 2489.07 (1.00) | 1028.33 (1.00) | -15.65 (1.00) | 62.88 (1.00) | 1280.29 (1.00) |
| PG+REVEAL-IT | **742.33 (0.90)** ↑ | 2253.70 (0.90) | 975.04 (0.90) | **-13.21 (0.90)** ↑ | **70.66 (0.90)** ↑ | **1546.38 (0.90)** ↑ |

Table 2: **REVEAL-IT improves learning efficiency on OpenAI GYM benchmark.** In this table, we report the evaluation performance of three RL algorithms and how the performance changes with REVEAL-IT. (-) indicates the environment steps (millions) that an agent takes in training. It's clear that REVEAL-IT makes it easier for RL agents to learn in several different environments.

**Comparison with other interpretability frameworks** In the experiments on ALFworld, we mainly focus on how REVEAL-IT can help in the RL training. However, it is necessary to evaluate the performance of our explanation mechanism, i.e., can other interpretability methods have better improvement for RL? Therefore, We add the comparison with other interpretability methods (GNNexplainer (Ying et al., 2019), MixupExplainer (Zhang et al., 2023)) and report the results in the Tab.ref 3. Due to the limited time, we can only compare with GNNexplainer and MixupExplainer in ALFworld, and we keep the hyperparameters the same as those in the released code.

## 5.3 Understanding the Relationship between Sub-tasks from Policy

This section will explore how the agent acquires knowledge in various sub-tasks from the perspective of policy. Given that the different nodes in the policy will be activated in the evaluation, how will

| Methods | Avg. | Pick | Clean | Heat | Cool | Look | Pick2 |
|---------|------|------|-------|------|------|------|-------|
| REVEAL-IT (ours) | **0.80** | **0.66** | **0.90** | **0.81** | **0.80** | **0.85** | **0.70** |
| REVEAL-IT (with GNNexplainer) | 0.64 | 0.47 | 0.70 | 0.61 | 0.69 | 0.70 | 0.55 |
| REVEAL-IT (with MixupExplainer) | 0.52 | 0.33 | 0.59 | 0.50 | 0.55 | 0.60 | 0.42 |

Table 3: **The comparison with other interpretability methods in REVEAL-IT.** In this ablation experiment, only the GNN explainer model was replaced, and other RL algorithms and parameters remained the same.

the weights linked to these nodes be modified? Can GNN-explainer learn to differentiate between the sections that have more significant updates in terms of weight? Fig. 2 illustrates updating the policy during training. Firstly, it is imperative to focus on the modifications in policy updates in every individual row. As training advances, we will observe that the sections with more significant policy updates will undergo modifications (shown by the thick lines). Furthermore, our findings indicate that as the training progresses toward the latter stage, there is a greater overlap between the region with a larger update amplitude and the region triggered by evaluation. This illustrates that GNN explainer comprehends the connection between policy training and evaluation. By carefully considering each vertical direction, we have identified and emphasized the sections of policies that are common to multiple activities (shown by gray boxes). By analyzing the neural network nodes within the gray box, we discovered that certain tasks, which necessitate agents to possess comparable abilities based on common knowledge, exhibit increasingly evident overlaps in their policies. For instance, the subtasks "open microwave 1" and "take apple 1 from microwave 1" both require the agent to comprehend the spatial location of the microwave (highlighted by the orange box). However, the second subtask also demands the agent identify the object "apple" and its specific location. There is less collaboration across distinct subtasks. Upon reviewing the prior distance, we observe that the subtask "put apple 1 on sinkbasin 1" shares some similarities with the previous subtask. Both subtasks are interconnected with "apple 1".

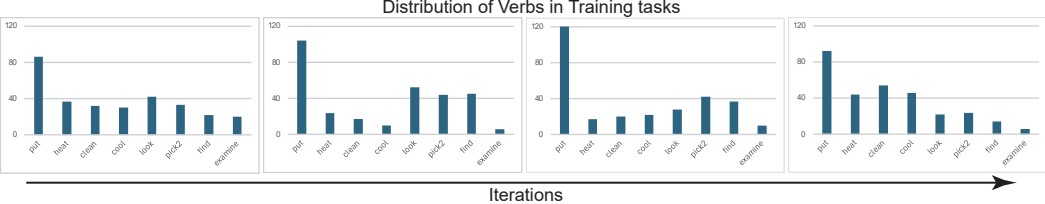

Figure 3: **The distribution of verbs in training tasks.** This figure demonstrates how REVEAL-IT optimizes the task sequences. We employ various verbs in tasks to differentiate, as each type necessitates distinct capabilities from the agent. The agent training process reflects the task sequence change from left to right. Analyzing the changes in task distribution shows that the "put" type of task is the most prevalent. As the training advances, the initial focus of training tasks is on teaching the agent to locate and retrieve objects in the environment ("look", "pick"). Subsequently, the agent is trained on tasks that require it to acquire other skills, e.g., "clean", "heat", and "examine". We provide a larger version in Appendix. E.

This finding demonstrates why our GNN-based explainer is capable of optimizing subtask sequences, i.e., the agent's learning process consists of both shared and distinct components. Once the agent becomes proficient at the intersecting components, decreasing the frequency of these specific training tasks is advisable. This is because the agent cannot acquire new knowledge from them, resulting in an improvement of the agent's training efficiency.

# 6 CONCLUSION AND FUTURE WORK

We present REVEAL-IT, a novel framework designed to explain the learning process of RL agents in complex environments and tasks while also serving as a foundation for optimizing task sequences. We demonstrate the mechanisms and functionalities of REVEAL-IT in complex environments. Nevertheless, REVEAL-IT does possess certain constraints. A primary constraint is its ability to adapt to multi-modal challenges. As an illustration, in the ALFWorld, we exclusively implemented the REVEAL-IT within the visual environment. Furthermore, it is worth investigating if REVEAL-IT can effectively transform the knowledge acquired from policies and training problems into natural language, enabling its broader and more direct application in other reinforcement learning domains.

SOCIETAL IMPACT

Our research centers on the explanation of an RL agent's training tasks and performance in a continuous control environment. This study aims to solve the problem of RL's explanation in complex environments and tasks. enhancing transparency and interpretability in RL. This framework can promote trust and understanding of AI decision-making processes by explaining an RL agent's training tasks and performance in complex environments. This increased transparency can lead to more responsible and ethical deployment of RL systems in various real-world applications, such as autonomous vehicles, healthcare diagnostics, and financial trading. Technological progressions possess the capability to augment industrial procedures, ameliorate efficacy, and mitigate hazards to human laborers across diverse domains.

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

# A VISUALIZED TASK EXAMPLES IN ALFWORLD

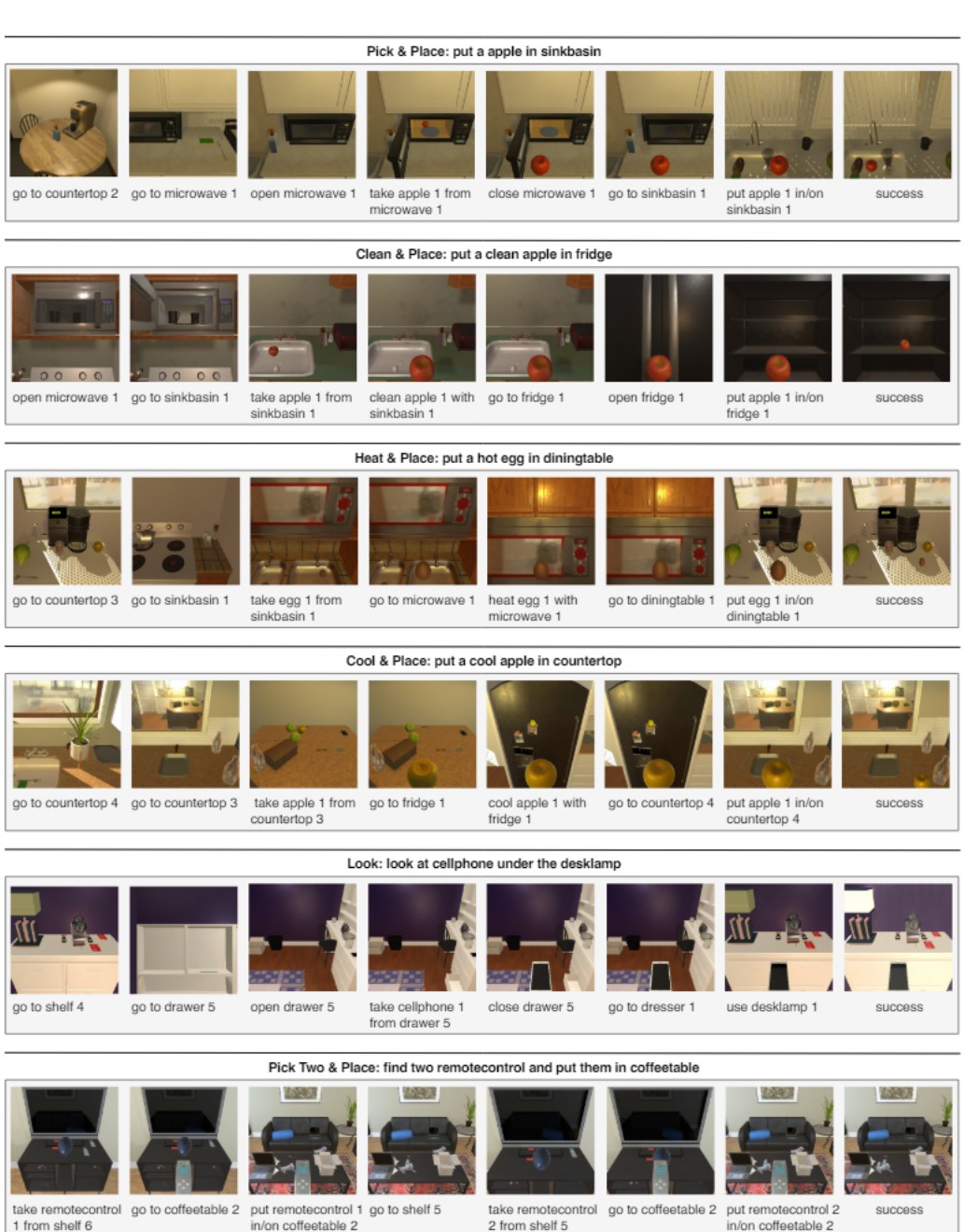

Figure 4: **Visualized task examples of ALFWorld** Shridhar et al. (2021). This benchmark utilizes various household scenarios created within the Ai2Thor environment. All objects can be moved to different locations based on available surfaces and class restrictions in this environment. This allows for the generation of a wide range of new tasks by combining different objects and goal positions in a procedural manner.

## B    THE DIFFERENCE BETWEEN REVEAL-IT AND OTHER EXPLANATION METHODS

In this section, We want to discuss the novelty of REVEAL-IT with other explanation methods in more detail, and we hope this discussion can help clarify the contribution of REVEAL-IT.

**An interactive node-link visualization of convolutional neural networks.** Note that utilizing the visualization to show the structure of a CNN/GNN network or an MLP network is previously discussed in Harley (2015). Nevertheless, the use of visualization is uncommon in RL, this is because:

- Gaining a visual representation of the policy structure does not enhance our comprehension of the possibilities of RL, and the motivation for visualizing CNNs is essentially distinct from this, i.e., we lack knowledge regarding the specific capabilities of the agent that correspond to a particular section of the policy. REVEAL-IT confirmed the accuracy of this mapping relationship for the first time (refer to Fig. 2).

- The process of updating policies necessitates millions of revisions. Without a proficient interpreter, meaningful conclusions cannot be derived from the visualization outcomes. The GNN explanation plays a crucial part in REVEAL-IT.

**The difference between Gnnexplainer/PGExplainer**: Generating explanations for graph neural networks". The primary objective behind the GNN explainer in REVEAL-IT is to identify the important policy updates that occur during the agent's learning process. This allows us to correlate these updates with the agent's abilities, thereby explaining how the agent learns in complex tasks. Simultaneously, the GNN explainer can assist us in determining if the predictor comprehends the inherent connection between the agent's abilities and the learning process, hence assessing the credibility of the task's significance. This differs radically from the conventional GNN explainer employed for identifying important edges.

**How does REVEAL-IT differ from other interpretable RL algorithms?** It is important to understand that REVEAL-IT stands apart from conventional interpretable RL algorithms in: (1) REVEAL-IT does not necessitate the use of artificially constructed causal structures. (2) REVEAL-IT does not require additional knowledge of SCM. These two points are the most significant limitations of classical explanation RL in complex and real environments. To overcome this flaw and restriction, we employ a "task-agent policy-GNN explainer" architecture, enabling RL use in complex environments and task scenarios. This architecture also comprehensively explains how RL agents learn in these tasks, explaining why an agent can succeed or fail after training.

## C    ADDITIONAL EXPERIMENTS BASED ON THE TEXT ENGINE IN AFLWORLD

To further evaluate the generalizability and adaptability of REVEAL-IT, we conducted a new set of experiments in the ALFWorld environment, focusing exclusively on interactions with the text engine. Unlike prior experiments that utilized the visual environment, these experiments aim to demonstrate the framework's capability to handle tasks requiring textual representations. We report the results in Table. 4.

| Agent | Success Rate | | | | | | |
|---|---|---|---|---|---|---|---|
| | Avg. | Pick | Clean | Heat | Cool | Look | Pick2 |
| ReAct (Yao et al., 2022) | 0.54 | 0.71 | 0.65 | 0.62 | 0.44 | 0.28 | 0.35 |
| AutoGen (Wu et al., 2023) | 0.77 | 0.92 | 0.74 | 0.78 | 0.86 | 0.83 | 0.41 |
| Reflextion (Shinn et al., 2023) | **0.91** | **0.96** | **1.00** | 0.81 | 0.83 | 0.94 | **0.88** |
| REVEAL-IT (Ours) | 0.86 | 0.72 | 0.96 | **0.87** | **0.82** | **0.95** | **0.90** |

Table 4: Comparison with text agents in ALFworld.

# D    MORE DETAILS ABOUT GNN EXPLAINER

## D.1    THE LOSS FUNCTION FOR GNN EXPLAINER

In REVEAL-IT, We use the loss in the PGEexplainer (Luo et al., 2020) to train the GNN explainer, and we use an MSE loss for the GNN predicter. On the other hand, in the traditional GNN explainer method, the learning objectives of the evaluator GNN and the explainer are often consistent, i.e., maximizing the mutual information between the label and the selected subgraph, and we can express this process as:

$$\max_{G_s} = MI(Y, G_S) = H(Y) - H(Y|G = G_S), \tag{4}$$

where $G_S$ denotes the masked subgraph, the GNN explainer optimizes this function locally by training an NN to provide a customized explanation for each instance. However, REVEAL-IT optimizes this by adopting an NN to learn the crucial edges on multiple instances (each evaluation results of RL).

## D.2    GNN EXPLAINER HIGHLIGHT THE IMPORTANT UPDATE

To extract the explanation subgraph $G_S$, we first compute the importance weights on edges. A threshold is used to remove low-weight edges and identify the explanation subgraph $G_S$. The ground truth explanations of all datasets are connected subgraphs. Therefore, we identify the explanation as the connected component containing the explained node in GS. We identify the explanation for the generated graph classification by the maximum connected component of $G_S$. For all basic RL algorithms, we perform a search to find the maximum threshold such that the explanation is at least of size $K_M$. When multiple edges have tied importance weights, all of them are included in the explanation.

## D.3    HOW TO ENSURE THE TRAINING PROCESS OF GNN+RL IS STABLE?

The fundamental principle of the GNN+RL training scheme is mutual learning, where the GNN predicts the value of the training task for the RL agent based on the RL training process, and the GNN predictor influences the training task of the RL agent. During the initial stages of training, the GNN predictor exhibits instability due to inadequate data. We employ a random sampling technique to select training tasks from the complete set. Once RL has accumulated a specific quantity of data, we systematically enhance the significance of the GNN predictor in choosing training tasks. Providing a precise theoretical guarantee for this procedure is challenging, yet it is a commonly used approach in curriculum RL to choose training tasks, e.g.,  Ao et al. (2023); Held et al. (2017).

## D.4    HOW DOES GNN EXPLAINER MAP THE SKILLS OF THE RL AGENT?

In REVEAL-IT, the central concern revolves around the GNN explainer's capacity to comprehend the agent's learning prcess and ascertain whether the agent has achieved the certain abilities in accomplishing the task. The GNN explainer is designed to identify and emphasize important updates in the policy. Suppose the nodes associated with these highlighted updates correspond to the nodes engaged by the agent during the evaluation. In that case, it can be inferred that the explanation is cognizant of the specific sections of the policy that align with the agent's essential abilities. Hence, we ascertain the ratio of nodes identified by the GNN explainer and nodes stimulated by the agent in successful tasks and report it in Fig. 5. In this series of comparisons, we incorporated a pre-trained Graph Neural Network (GNN) explanation that utilizes data from RL training data from different sequences of training tasks in the same environment. An intriguing observation is that we discovered that the efficacy of the GNN-explainer, when trained on various tasks, was inferior than that of the GNN-explainer taught using a reinforcement learning agent. This discrepancy may arise from the varying demands that different tasks place on the training process of reinforcement learning (RL) agents. Additionally, pre-trained graph neural networks (GNNs) may struggle to comprehend the ongoing RL training progress accurately due to the incorporation of a priori bias.

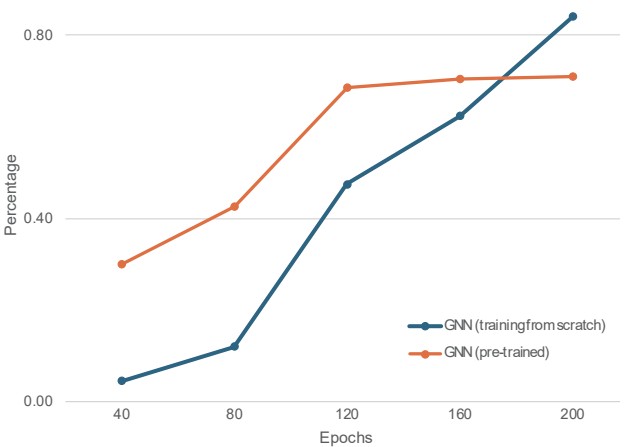

Figure 5: **The results of learning process of the GNN explainer.**

# E    LARGE VERSION OF FIGURES IN THE MAIN PAPER

 Due to the limited space of the main paper, the figures in the main paper might not be clear enough to read. We provide a lager version here for reading.

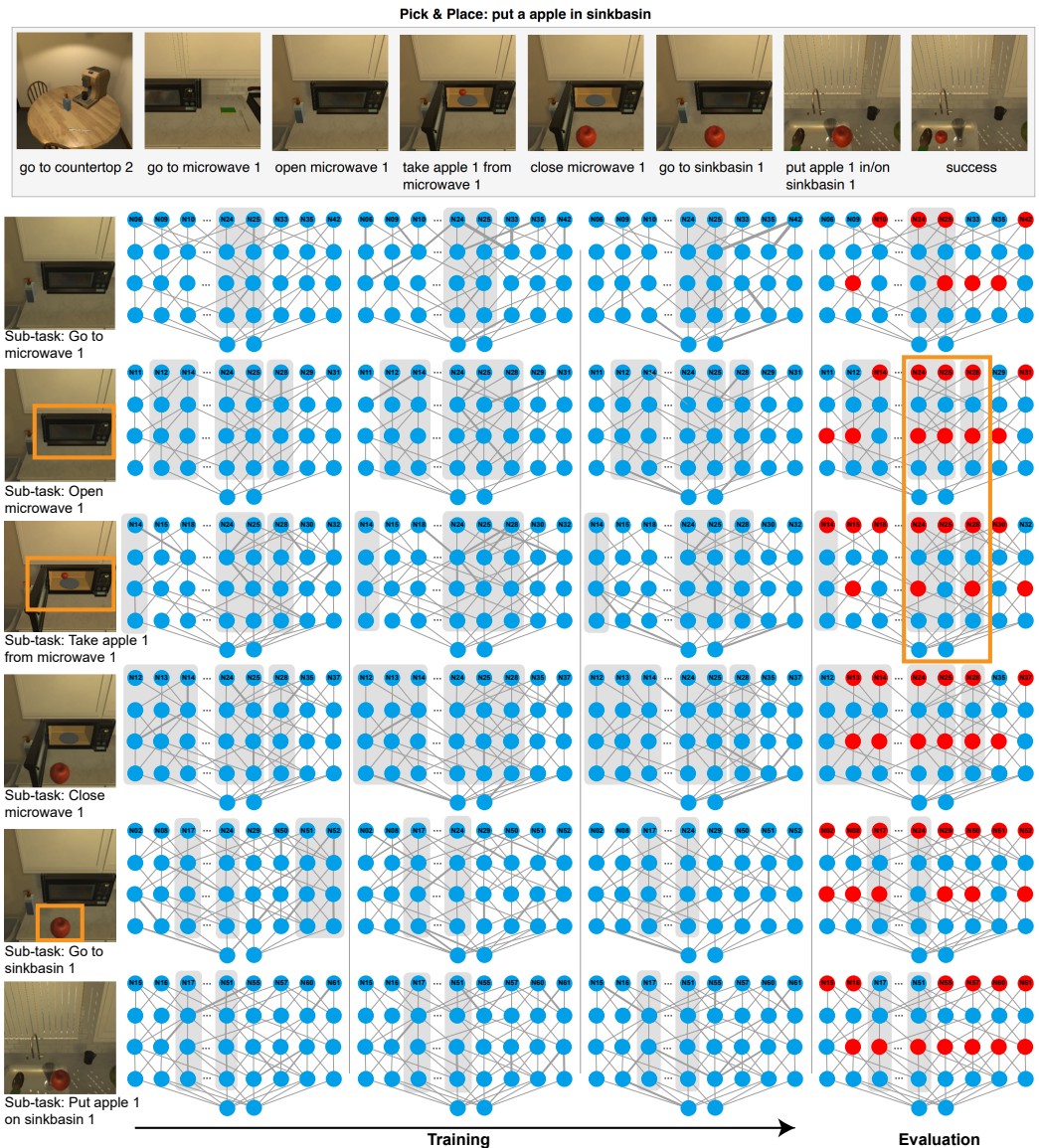

Figure 6: **Visualized important policy updates by GNN explainer.** In this figure, the first line depicts the sequential arrangement of sub-tasks that must be accomplished step-by-step to accomplish a given task fully. One of the sub-tasks is located on the left side of the second line. The first to third columns of the tree diagram illustrate the RL policy update process. The blue circles represent nodes in the neural network, while the connections between the circles represent weight updates. Thicker connections indicate larger updates in weight amplitude (selected by GNN explainer). The red circles in the tree diagram on the far right illustrate the active policy nodes during the evaluation process. The links here represent the revised weights throughout the training phase of this subtask. The orange squares indicate the portions of the policy that are common to several sub-tasks. We opt to depict the 8 interconnected nodes with the most significant weight adjustment, facilitating comprehension of the reinforcement learning policy's learning process and highlighting the policy's shared component more distinctly.

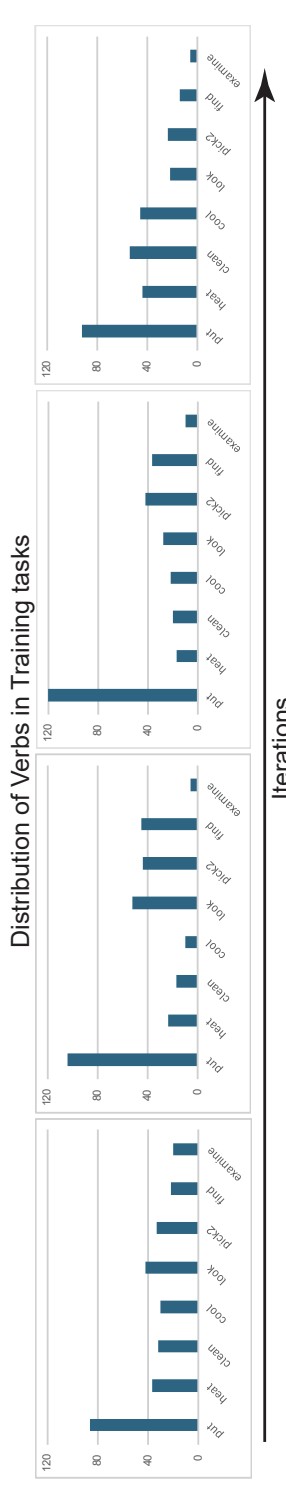

Figure 7: **The distribution of verbs in training tasks.** This figure demonstrates how REVEAL-IT optimizes the task sequences. We employ various verbs in tasks to differentiate, as each type necessitates distinct capabilities from the agent. The agent training process reflects the task sequence change from left to right. Analyzing the changes in task distribution shows that the "put" type of task is the most prevalent. As the training advances, the initial focus of training tasks is on teaching the agent to locate and retrieve objects in the environment ("look", "pick"). Subsequently, the agent is trained on tasks that require it to acquire other skills, e.g., "clean", "heat", and "examine".

