# OpenReview forum: "REVEAL-IT: REinforcement learning with Visibility of Evolving Agent poLicy for InTerpretability"
_ICLR.cc/2025/Conference — Submitted to ICLR 2025_

### Official Review · Reviewer_Lyow · 2024-10-24

**Soundness:** 2
**Presentation:** 2
**Contribution:** 2
**Rating:** 5
**Confidence:** 3

**Summary:**

The authors introduce an interpretable RL agent, in the context of environments with sub-tasks. The algorithm uses a GNN-based approach to visualise the updates to the policy, and also uses another GNN to implement curriculum learning.

**Strengths:**

The authors introduce a new framework for RL interpretability which can provide better insights than other methods. The method also includes a curriculum learning component, which produces very strong results on the ALFWorld environment. The authors do also spend some time analysing the results from their interpretability framework, in order to showcase its capabilities.

**Weaknesses:**

The authors provide a valid criticism of more generic interpretability methods e.g. post-hoc methods like saliency maps. However, this work assumes (unless I am wrong - see below) the environment provides a set of subtasks that can be trained on, which is a large assumption. Therefore the generality of this method is somewhat limited. To what extent do subtasks need to be provided/can be inferred?

The methodology is a bit unclear, as the focus appears to be on interpretability, and then halfway through the paper the authors introduce a GNN predictor and curriculum learning. Curriculum learning is not discussed at all in the Related Works section. And yet the authors show that their method significantly outperforms other methods in ALFWorld (Table 1), so it is clear that this is not just about interpretability. If this is the case, then the authors should be mentioning this in the abstract and from the introduction.

I am also under the impression that environment subtasks are needed for REVEAL-IT, but the authors perform experiments on OpenAI Gym MuJoCo environments, which don't have them? Table 2 indicates that adding REVEAL-IT to existing methods improves their performance, but in Table 1 the authors present REVEAL-IT as its own algorithm, so once again it is unclear what is going on here. Is REVEAL-IT standalone or an addition to existing algorithms?

The paper should be checked for spelling mistakes, e.g., "Strucutral" on page 4.

**Questions:**

In general the authors need to be clearer about the system they have developed, and if curriculum learning is indeed part of the system, it should be discussed more thoroughly and introduced early on.

Also, very little is said about the Gym tasks, so it is difficult to understand what went on in those experiments, particularly as the standard environments do not have subtasks.

---

> ### Author Response · Authors · 2024-11-22
> **Thank you for the review! We hope our reply can address your concerns!**
>
> Thank you for your time and valuable suggestions! We have updated the paper following your suggestions; please refer to the highlighted part in the rebuttal revision. Now, we address your main concerns as below:
>
> **REVEAL-IT does not require for stronger assumptions than other interpretability methods.**
> > However, this work assumes (unless I am wrong - see below) the environment provides a set of subtasks that can be trained on, which is a large assumption. Therefore the generality of this method is somewhat limited. To what extent do subtasks need to be provided/can be inferred?
> - The The main reason that the traditional conterfacutal/causal methods can't work in complex environments/tasks is that constructing a true causal model/SCM relies on a perfect understanding of the data generation process, any correlation, intervention and even counterfactual inquiries. However, given the inherent complexity of the world, it is often impractical to access a fully specified SCM. This claim has been fully discussed in previous works and survey papers[1], [2].
> [1] Bernhard Schölkopf, Francesco Locatello, Stefan Bauer, Nan Rosemary Ke, Nal Kalchbrenner, Anirudh Goyal, and Yoshua Bengio. Toward causal representation learning.
> [2] Jean Kaddour, Aengus Lynch, Qi Liu, Matt J. Kusner, and Ricardo Silva. Causal machine learning: A survey and open problems.
>
> **What if the environment/task does not have apparent sub-task structure for training?**
> > I am also under the impression that environment subtasks are needed for REVEAL-IT, but the authors perform experiments on OpenAI Gym MuJoCo environments, which don't have them?
> - Regarding to the GYM environment, we agree that there are no obvious sub-tasks in GYM or similar environments. Instead of using a random sample from the replay buffer (PPO or other online RL methods), we store the trajectories that bring the agent higher learning progress and do the experience replay based on these trajectories (similar to the idea in HER[1]).
>
> [1] Marcin Andrychowicz, Filip Wolski, Alex Ray, Jonas Schneider, Rachel Fong, Peter Welinder, Bob McGrew, Josh Tobin, Pieter Abbeel, Wojciech Zaremba, Hindsight Experience Replay, NIPS 2017.
>
> **Is REVEAL-IT an addition to existing algorithm?**
> > Table 2 indicates that adding REVEAL-IT to existing methods improves their performance, but in Table 1 the authors present REVEAL-IT as its own algorithm, so once again it is unclear what is going on here. Is REVEAL-IT standalone or an addition to existing algorithms?
> - In terms of the structure of our method, we can say REVEAL-IT is an addition to existing algorithms. However, REVEAL-IT does not have any constraints on the basic RL algorithm used for the control policy training. This is the reason we compare REVEAL-IT (PPO-based) with different baselines in ALFworld, and we compare with classical RL algorithms in GYM.
>
> **Curriculum learning is not the key contribution.**
> > The methodology is a bit unclear ... and then halfway through the paper the authors introduce a GNN predictor and curriculum learning. Curriculum learning is not discussed at all in the Related Works section. And yet the authors show that their method significantly outperforms other methods in ALFWorld (Table 1), so it is clear that this is not just about interpretability.
> - REVEAL-IT does not exclusively focus on interpretability but acts as a framework that improves the understanding of an RL agent's learning process and improves the agent's learning efficiency based on it. These two contributions are complementary rather than contradictory.
> - We do not claim to innovate in curriculum learning itself but use it to demonstrate the utility of its explanation-driven approach. Curriculum learning is a tool within REVEAL-IT to operationalize the insights derived from the GNN explainer. We have clarified this in Line.163-166.
> - Although curriculum learning is not our main contribution, we agree that including the discussion in related work can improve the paper. Thank you for the suggestion. Please refer to the new vision for the udpates.

---

> > ### Comment · Reviewer_Lyow · 2024-11-24
> > **Clarifying Questions**
> >
> > Thank you for the responses and updates to the paper.
> >
> > > REVEAL-IT does not require for stronger assumptions than other interpretability methods.
> >
> > REVEAL-IT does require an environment with sub-tasks, or at least a way to split the data? Some interpretability methods, such as saliency maps, do not require this, but _causal_ interpretability methods have more requirements?
> >
> > > What if the environment/task does not have apparent sub-task structure for training?
> >
> > In the case of the MuJoCo environments, does REVEAL-IT bring any interpretability insights?
> >
> > > Is REVEAL-IT an addition to existing algorithm?
> >
> > In that case it would be clearer to refer to "REVEAL-IT" in Table 1 as "PPO+REVEAL-IT", as in Table 2?
> >
> > > Curriculum learning is not the key contribution.
> >
> > Understood - the new subsection in Related Works makes REVEAL-IT's relationship to this field clearer.

---

> > > ### Author Response · Authors · 2024-11-25
> > > **Thank you for the response. We are glad to hear that some of your main concerns have been addressed.**
> > >
> > > Thank you for the response. We are glad to know that our rebuttal has addressed your main concerns! Here is our reply to your remained concerns:
> > >
> > > **REVEAL-IT does not have the requirements of strong assumptions.**
> > > > REVEAL-IT does require an environment with sub-tasks, or at least a way to split the data? Some interpretability methods, such as saliency maps, do not require this, but causal interpretability methods have more requirements?
> > > - No, this is not true. As we mentioned in the rebuttal, we store the trajectories with higher improvement for experience replay in the environment without obvious sub-task structure. Moreover, we don't think this can be categorized into the method of "split the data" since we still use the whole replay buffer for training. For example, in the context of goal-conditioned RL, the MDP tuples in the replay buffer include the goal information ($\mathcal B =\{(s_t,a_t,r_t,g_n)\}$, where $g_n$ denotes a goal state in goal space $\mathcal G$). We don't regard the goal-conditioned policy as trained on splitting the data.
> > >
> > > **REVEAL-IT in Mujoco.**
> > > > In the case of the MuJoCo environments, does REVEAL-IT bring any interpretability insights?
> > > - First, we want to emphasize that the targeting problem of REVEAL-IT is to provide explanations for the agent learning process in complex environments and tasks. The policy structure visualization can be easy for humans to understand based on the task description. We include the experiments in the OpenAI GYM environment to show that REVEAL-IT can also work/bring improvment to the basic RL algorithms in general environments.
> > > - We have conducted new experiments per your request in mujoco. Due to the limited time, we can only compare with two baselines, and we will include the whole benchmark in the future. Here are the results:
> > >
> > > |Agent| Ant-v3 |Swimmer-v3 | Hopper-v3 | HalfCheetah-v3|
> > > |:--|:--|:--|:--|:--|
> > > |REVEAL-IT+PPO|2745.57 $\pm$ 564.23|340.58 $\pm$ 6.20|2167.90 $\pm$ 102.81|6047.82 $\pm$ 87.21|
> > > |PPO|1480.47 $\pm$ 407.39|281.78 $\pm$ 11.86|2410.11 $\pm$ 9.86|5836.27 $\pm$ 171.68|
> > > |A2C|-15.93 $\pm$ 6.74|199.91 $\pm$ 1.32|679.01 $\pm$ 302.76|3096.61 $\pm$ 82.49|
> > >
> > > Please let us know if our response has addressed your concerns, and we are looking forward to having further discussions. We appreciate your valuable feedback and thoughtful suggestions, which have significantly helped us improve the clarity, depth, and presentation of our work. We sincerely hope you can consider kindly raising the score given that we've addressed all your comments.
> > > ****

---

### Official Review · Reviewer_tFxe · 2024-11-02

**Soundness:** 2
**Presentation:** 2
**Contribution:** 2
**Rating:** 5
**Confidence:** 4

**Summary:**

This paper proposes a GNN-based explainer to identify critical nodes or components for reinforcement learning (RL) tasks, with the goal of improving interpretability and enhancing the learning efficiency of RL agents. The approach involves visualizing a node graph that represents the RL training process for sub-tasks, and training the GNN-based explainer to approximate the true learning process in order to identify important components (key weights or edges) for the tasks. The GNN explainer is then used to guide policy learning. Results show improvements over standard RL models and language-based approaches (tested on ALFworld and other RL benchmarks).

Overall, the paper presents an interesting direction by learning critical components across multiple RL tasks through policy network responses. However, some technical aspects of the method are unclear and could benefit from further justification and improvement. I have outlined specific questions and concerns below and will give a borderline reject in this initial review.

**Strengths:**

- **[Motivation]**: The motivation is generally sound; learning from policy behavior appears to be a promising approach for developing interpretable and generalizable policies in complex environments.

- **[Empirical Evaluation]**: The empirical evaluation is relatively comprehensive, and the reported results show the potential of this approach.

**Weaknesses:**

**[About Problem Definition, Methodology, and Experiments:]**
- 1. The authors assert that variability in learning across models will not pose an issue since the learned explainer is model- or task-specific. However, for applications in multi-task learning, pre-training, or other forms of generalization, this variability is crucial and challenging. For instance, training the same network multiple times may yield high variance in the training process data due to permutation invariance. Empirical evaluation or theoretical justification for this would be useful.

- 2. To ensure the framework is generalizable and learns universal, principled representations, it would be beneficial to further explore the alignment between the learned structural information and the actual policies or concepts, either empirically or theoretically. Approaches could include using sparse autoencoders (potentially with larger models) [1] or examining the alignment between individual components and their corresponding concepts, modularities, interactions, and causal relations [2-5].

- 3. Building on point 2, utilizing these representations could facilitate compositional and hierarchical structures in policy adaptation and generalization. Including evaluations that focus on different levels of generalization would be uesful.

- 4. The impact of network size on the results should be investigated through ablation studies. If the network size is small, do the same phenomena observed in Figure 2 still occur?

- 5. What are the primary benefits of using GNNs for contribution analysis of each node or weight? Why not directly use magnitude, partial derivatives, or conditional mutual information to assess the importance of each weight?

**[About Clarity]**

- 1. It would be helpful to list all objective functions in a separate subsection, particularly the objectives for the GNN predictor and explainers, along with an explanation of how guidance information is provided for policy updates.

- 2. In line 115, the process is mentioned as being similar to a POMDP; please formulate this for clarity.

- 3. There are some typos to address, such as in line 11 of Algorithm 1—should $\pi_0$ be $\pi_t$? Also, in line 432, figure 4 should likely be referenced as figure 3 instead.

**Others**: As a side note, many causal RL works focus on learning world models, akin to a subgroup of model-based RL with interventions, rather than behaviors or policies/reward structures, which differ from the goals of this paper. The authors mention their inability to handle complicated tasks, but a more justified statement regarding this limitation should be provided.




[1] Gao, Leo, et al. "Scaling and evaluating sparse autoencoders." arXiv preprint arXiv:2406.04093 (2024).

[2] Marks, Samuel, et al. "Sparse feature circuits: Discovering and editing interpretable causal graphs in language models." arXiv preprint arXiv:2403.19647 (2024).

[3] Gandikota, Rohit, et al. "Erasing Conceptual Knowledge from Language Models." arXiv preprint arXiv:2410.02760 (2024).

[4] Geiger, Atticus, et al. "Causal abstraction: A theoretical foundation for mechanistic interpretability." Preprint (2024).

[5] Lippe, Phillip, et al. "Biscuit: Causal representation learning from binary interactions." Uncertainty in Artificial Intelligence. PMLR, 2023.

**Questions:**

I listed the questions and suggestions together with weaknesses in the above section.

---

> ### Author Response · Authors · 2024-11-22
> **Thank you for the review! We hope our reply can address your concerns!**
>
> Thank you for your time and valuable suggestions! We have updated the paper following your suggestions, please refer to the highlighted part in the rebuttal revision. Now, we address your main concerns as below:
>
> **REVEAL-IT is not trained across models.**
> > The authors assert that variability in learning across models will not pose an issue since the learned explainer is model- or task-specific. However, for applications in multi-task learning, pre-training, or other forms of generalization, this variability is crucial and challenging.
> - REVEAL-IT is purposefully designed to address specific tasks and environments rather than pre-trained setups. On the other hand, ALFworld is a benchmark that contains different tasks (refer to the task list in Appendix A). Therefore, the specificity is a feature, not a limitation, ensuring tailored explanations and efficient training for a given task or model.
> - REVEAL-IT does not have any specific requirements for the basic RL algorithm for training the control policy, which is not equal to REVEAL-IT can work with different RL algorithms in different environments/tasks simultaneously.
>
> **REVEAL-IT can work with simple networks, but it does not target on this.**
> > The impact of network size on the results should be investigated through ablation studies. If the network size is small, do the same phenomena observed in Figure 2 still occur?
> - If the RL policy's structure is overly simplistic, i.e., every node can be completely active during the testing phase, the role of REVEAL-IT may be constrained. However, generally speaking, a simple policy typically correlates to a simple job and environment, which is not the aim that REVEAL-IT needs to solve.
> - On the other hand, the structure of RL policy in GYM is simpler than it in ALFworld. The results in Table 2 prove that REVEAL-IT can still bring some improvement in some tasks.
>
> **The benefits of GNN explainer.**
> > What are the primary benefits of using GNNs for contribution analysis of each node or weight? Why not directly use magnitude, partial derivatives, or conditional mutual information to assess the importance of each weight?
> - REVEAL-IT is designed to intuitively visualize the learning process of the agent and help people understand what specific abilities the agent has learned in a subtask to enable it to complete the final task. Therefore, we choose to visualize the policy update process, but considering the complex structure of the policy itself and the excessive number of updates. Even with the visualization, the results are not easy to understand; thus, we design a GNN explainer to simplify the policy graph and highlight important updates.
>
> **The gap between the learned structural information and the policy has already been addressed.**
> > To ensure the framework is generalizable and learns universal, principled representations, it would be beneficial to further explore the alignment between the learned structural information and the actual policies or concepts, either empirically or theoretically.
> - The GNN explainer in REVEAL-IT is trained on the visualized graphs of the control policy updates. Meanwhile, the experiment results in ALFWorld demonstrate that the framework can identify and utilize key updates to optimize task sequencing, which indirectly validates the alignment between learned structures and policies.
>
> **Compare with other encoder methods.**
> > Approaches could include using sparse autoencoders (potentially with larger models) [1] or examining the alignment between individual components and their corresponding concepts, modularities, interactions, and causal relations [2-5].
> - In ALFworld, REVEAL-IT is trained and tested in the visual environment, which has no prior/extral knowledge from the text engine (text world in ALFworld). This is the same for the baseline methods (BLIP-2, LLaMA-Adapter, InstructBLIP and MiniGPT-4.)
> - To compare with other encoder methods, we need to train REVEAL-IT in the text engine (without visual environment), and also we need to compare it with the baselines in the same conditions. Therefore, we have added new experiments for comparison with LLM-based baselines. We report the results in the table below, and we have also included it in the paper. Please note that the experiment setting is different with the conditions in Table 1.
>
> |Methods| Avg.|Pick|Clean|Heat|Cool|Look|Pick2|
> |:--|:--|:--|:--|:--|:--|:--|:--|
> |REVEAL-IT| 0.86|0.72|0.96|**0.87**|**0.82**|**0.95**|**0.90**|
> |ReAct|0.54|0.71|0.65|0.62|0.44|0.28|0.35|
> |AutoGen|0.77|0.92|0.74|0.78|0.86|0.83|0.41|
> |Reflextion|**0.91**|**0.96**|**1.00**|0.79|**0.82**|0.94|0.88|
>
> **ReAct:** Shunyu, et al. React: Synergizing reasoning and acting in language models. In ICLR, 2023.
> **Reflextion:** Noah Shinn, et al. Reflexion: Language agents with verbal reinforcement learning. In NeurIPS, 2023.
> **AutoGen:** Qingyun Wu, et al. Autogen: Enabling next-gen llm applications via multi-agent conversation framework.

---

> ### Comment · Reviewer_tFxe · 2024-11-22
>
> First of all, I would like to thank the authors for their detailed response. I still have a few questions that I would appreciate further clarification on.
>
> - For Q1, my question is whether training the same model (using essentially identical policy architectures) multiple times would yield consistent interpretability results. How do you ensure alignment of the neural representations in this case? Perhaps I am misunderstanding something, but further clarification would be greatly appreciated.
>
>
>
>
>
> - Regarding the question I raised about the "alignment between the learned structural information and the actual policies or concepts," I did not find a specific answer to this. Could the authors provide some clarifications or pointers?
>
>
> - Regarding this one:
>
> > REVEAL-IT is designed to intuitively visualize the learning process of the agent and help people understand what specific abilities the agent has learned in a subtask to enable it to complete the final task. Therefore, we choose to visualize the policy update process, but considering the complex structure of the policy itself and the excessive number of updates. Even with the visualization, the results are not easy to understand; thus, we design a GNN explainer to simplify the policy graph and highlight important updates.
>
> I understand your point here, but what confuses me is whether there is any fundamental evidence demonstrating that this approach is inherently better than alternatives, such as using magnitude, partial derivatives, or conditional mutual information, which I believe can also be used for visualization.
>
> - Side note: Thank you for considering and comparing additional encoders.  Very impressive and helpful

---

> > ### Author Response · Authors · 2024-11-23
> > **Thank you for the reponse. We provide detailed explanations on your concerns.**
> >
> > Thanks for your time, and we appreciate your feedback. We are glad to hear your acknowledgement of our additional experiments. We believe your suggestions will provide significant improvement to our paper and more value to our research in the future. Here is the reply to your question:
> >
> > **REVEAL-IT is stable.**
> > > For Q1, my question is whether training the same model (using essentially identical policy architectures) multiple times would yield consistent interpretability results.
> > - In all experiments in our paper, the reported results are the averaged performance. In ALFworld, we run the agent in 134 AlfWorld environments across six different tasks. In GYM environment, we have 6 random seeds. Due to the limited space, we didn't include std. in table 2.
> >
> > **To our knowledge, we have no alignment issues.**
> > > Regarding the question I raised about the "alignment between the learned structural information and the actual policies or concepts," I did not find a specific answer to this. Could the authors provide some clarifications or pointers?
> >
> > - In our rebuttal, we claimed that the training data used for the GNN explainer is exactly **the updates in the RL policy,** which serves as a controller to the agent. Therefore, we do not think we have an alignment issue between the GNN explainer and the actual policy.
> > - If possible, we hope you can provide more clarification on the "alignment between learned structural information and the actual policies" to let us know if there is any misunderstanding.
> >
> > **GNN Explainer has irreplaceable benefits.**
> > > what confuses me is whether there is any fundamental evidence demonstrating that this approach is inherently better than alternatives, such as using magnitude, partial derivatives, or conditional mutual information, which I believe can also be used for visualization.
> > - We cannot agree that the mentioned methods can replace GNN in REVEAL-IT.
> > - Magnitude-Based methods which focus on the size of weight updates assume that larger updates inherently signify greater importance. However, **in RL, the utility of an update is often context-dependent**, i.e., large updates during early training might indicate instability rather than meaningful learning, while small updates in a fine-tuning phase could have disproportionate effects on performance. Meanwhile, magnitude-based methods cannot distinguish between updates that improve task-specific performance and those that result from noise or unrelated factors. Previous works [1, 2] show that gradient noise in RL algorithms can lead to misleading magnitudes in updates (e.g., noisy gradients from policy gradients or exploration-driven updates in Q-learning).
> > - Partial derivatives methods will be limited to local sensitivity and non-compositionality. Partial derivatives focus on the immediate sensitivity of outputs to inputs, providing a local view of importance. **In RL, this fails to capture the global dependencies across policy updates or the multi-step nature of decision-making**. More specifically, partial derivatives are **less effective in tasks where policies involve long-term dependencies** (e.g., ALFWorld tasks requiring sequential planning). On the other hand, the RL environment exhibits delayed rewards, making it hard to correlate local sensitivities with task outcomes.
> > - CMI-based methods require estimating the mutual information between inputs and outputs conditioned on subsets of the data or model parameters. In environments like ALFWorld, where state representations are high-dimensional and tasks require adaptive policy updates, **computing CMI for every policy component would be prohibitively expensive and prone to estimation errors**. CMI assumes the availability of sufficient and stationary data, which is **rarely the case in RL due to exploration and environment variability** (state-action spaces are large and require extensive sampling to accurately compute distributions).
> >
> > [1] Pierluca D’Oro, Wojciech Jaskowski, How to Learn a Useful Critic? Model-based Action-Gradient-Estimator Policy Optimization, NIPS 2020.
> >
> > [2] Sutton, R. S., McAllester, D. A., Singh, S. P., & Mansour, Y., Policy Gradient Methods for Reinforcement Learning with Function Approximation.
> >
> > We hope this reply will help in addressing your concerns. Thank you agian for your valuable time and discussion.

---

> > > ### Comment · Reviewer_tFxe · 2024-11-23
> > >
> > > Thank you for your further clarification. I now have a better understanding of the work. I recommend that the authors include these discussions in future revisions of the paper. I will keep an eye on the discussions between the authors and other reviewers and make any necessary changes to my recommendation or rating accordingly.

---

### Official Review · Reviewer_jsXE · 2024-11-04

**Soundness:** 2
**Presentation:** 1
**Contribution:** 2
**Rating:** 3
**Confidence:** 2

**Summary:**

The authors propose a framework for interpreting the training process of RL algorithms. Policy updates are visualized with node-link graphs, where the nodes are the neurons in the policy network and the edges are the weights that were updated. A GNN predictor is then trained to predict the RL algorithm’s learning progress, defined as the increase in return on a task after one policy update. A GNN explainer is trained to find which updated weights are most critical for the success of the RL agent, by finding the subset of weights that preserves the GNN predictor’s output given only that subset. The authors demonstrate that REVEAL-IT's explanations can be used to improve training efficiency and performance of various RL algorithms in ALFWorld and OpenAI gym environments.

**Strengths:**

REVEAL-IT addresses an important challenge in deep RL.

The method is broadly applicable, as it is agnostic to the environment or (online) RL algorithm.

The performance appears to be quite impressive for Alfworld.

**Weaknesses:**

The writing is very difficult to follow due to excessive verbosity, vague language and grammar issues e.g. “you can record and correspond to the changes in the value of a specific part of the weights” (line 186) or “the understanding the critical nodes in the RL agent’s evaluation is a crucial pre-requisite for determining the significance of weights updating.” (line 220) The authors should review the paper for conciseness and grammatical accuracy

The GNN Explainer does not seem to provide much human-interpretability. Figure 2: “we will observe that the sections with more significant policy updates will undergo modifications” seems to be a trivial observation rather than something illuminating. It is not uncommon for deep RL models to have millions of weights, so the ability to highlight a subgraph of most important weight updates would still leave the user with far too many to interpret. Could the authors provide concrete examples of helpful insights gained from the GNN Explainer?

Results tables are missing standard deviations; particularly for Table 2 it is unclear whether the improvements are significant.

The link to view the project on line 311 is broken.

Some figure references are broken, e.g. the distribution of training subtasks is figure 3 but referenced as figure 4 on line 430

**Questions:**

What is the GNN Explainer’s training objective? If it is only trained to preserve the predictor’s accuracy, then it could just output the full graph.

What is the definition of “active nodes” in Step 1 of section 4.2?

How exactly does the GNN explainer choose the distribution of subtasks to train on? That does not seem to be a direct byproduct of classifying the most critical weight updates. And how does it help on the OpenAI gym environments which do not involve any subtasks?

The conclusion states that REVEAL-IT can’t adapt to multi-modal challenges. Why would it not be able to handle non-visual modalities? It seems like it can be applied wherever a neural network policy network is used, which does not seem to be constrained to image inputs.


In Figure 2, what do the gray shaded regions correspond to? “Thicker connections indicate larger updates in weight amplitude (selected by GNN explainer)” - does this mean that thicker connections indicate weights that were both selected by GNN explainer, and had large updates in amplitude? How were the portions of the policy that are common to several sub-tasks identified? “as the training progresses toward the latter stage, there is a greater overlap between the region with a larger update amplitude and the region triggered by evaluation.” - what does “the region triggered by evaluation” refer to?

---

> ### Author Response · Authors · 2024-11-22
> **Thank you for the review! We hope our reply can address your concerns!**
>
> Thank you for your time and valuable suggestions! We have updated the paper following your suggestions, please refer to the highlighted part in the rebuttal revision. Now, we address your main concerns as below:
>
> **The definition of activated nodes in RL policy.**
> > Question 1&2: What is the GNN Explainer’s training objective? What is the definition of “active nodes” in Step 1 of section 4.2?
> - We add a ReLU function on the first and third layers in the policy network to visually identify the activated nodes through the evaluation process. We introduced this in section 5.2 line 352-355.
> - The GNN explainer learns to simplify the visualization of policy updates to help humans understand in which sub-tasks the agent has learned relevant capabilities to complete the task. Please refer to the weakness.4 in our reply to reviewer mUUC for more details of GNN explainer.
> - The GNN explainer is trained to simplify the original graph (policy visualization) **to a simpler sub-graph that the GNN predictor can still make the same prediction on it.**
>
> **The sub-task sequence optimization in REVEAL-IT.**
> > How exactly does the GNN explainer choose the distribution of subtasks to train on? and how does it help in GYM environment?
> - GNN explainer does not directly change the distribution of subtasks for training. Alternatively, we rank the sub-tasks in terms of the learning progress predicted by the GNN predictor. We introduce this in section 4.2 Line 247-251. To avoid falling into the local optimum and the situation where the predictor is not accurate enough in the early stage of training, we introduced $\epsilon-$greedy in task selection (refer to Line4 in Alg.1).
> - Regarding to the GYM environment, we agree that there are no obvious sub-tasks in GYM or similar environments. Instead of using random sample from the replay buffer (PPO or other online RL methods), we store the trajectories that bring the agent higher learning progress and do the experience replay based on these trajectories (similar to the idea in HER[1]).
>
> [1] Marcin Andrychowicz, Filip Wolski, Alex Ray, Jonas Schneider, Rachel Fong, Peter Welinder, Bob McGrew, Josh Tobin, Pieter Abbeel, Wojciech Zaremba, Hindsight Experience Replay, NIPS 2017.
>
>
> **GNN explainer does not choose the sub-tasks for training.**
> > What is the GNN Explainer’s training objective? If it is only trained to preserve the predictor’s accuracy, then it could just output the full graph.
> > How exactly does the GNN explainer choose the distribution of subtasks to train on?
> - We have already provided learning objectives for the GNN explainer in Appendix C in the original edition. And now, we move this part to Line.310-320. The purpose of the GNN explainer is to prune off unimportant edges (unimportant updates in policy). Due to space limitations and the fact that this is not much different from other GNN explainer methods from a technical perspective, we put this part in the appendix.
> - We store the visualized policy update information and the corresponding RL evaluation progress into a buffer, and use samples from this buffer to train the GNN explainer. Therefore, GNN explainer does not need to choose the sub-tasks.
>
> **Multi-modal challenges.**
> > The conclusion states that REVEAL-IT can’t adapt to multi-modal challenges.
> - The structure and environment of RL policy are not restricted by REVEAL-IT. In theory, REVEAL-IT should be applicable to multi-modal challenges. However, we find this could make the paper presentation more complex. For instance, in ALFworld, two distinct policies are necessary to interact with the visual engine and the text engine. We are of the opinion that the problem will be overcomplicated if REVEAL-IT is employed in multiple policies/agents tasks. We are extremely appreciative of the valuable suggestions you have provided, as they are of significant importance to our future research. We also concur that including such a statement directly in the conclusion would likely lead to reader confusion, and we have made the necessary adjustments.
>
> **More explanations on Figure 2.**
>
> - **The gray shaded regions.** The gray shaded regions in Fig.2 represent the shared nodes that have significant updates across different sub-tasks. Please note the node number in Fig.2, we understand that the number could be too small to read; there is a larger version in the appendix.
> - **Do thicker connections indicate weights that were both selected by GNN explainer and had large updates in amplitude?** Yes.
> - **the region triggered by evaluation**. We have added a ReLU function in the first and third layers in the control policy so that we can check which nodes are activated during the evaluation phase.
>
> **The link to view the project on line 311 is broken.**
> - We checked the link and found it can work. Maybe this is caused by the error of the website.

---

> > ### Author Response · Authors · 2024-11-25
> > **Thank you for your effort to ICLR! We are looking forward to hear from you.**
> >
> > Thank you again for your valuable review of our submission. We have carefully considered your comments and incorporated responses to address the concerns raised. In particular, we clarify the sub-task-related issues in REVEAL-IT, and we have improved the unclear parts in the paper. We hope our rebuttal provides sufficient detail to address your points effectively.
> >
> > We have also conducted new experiments per the reviewer tFxe and Lyow's requests. We hope the new experiments can help to strengthen your understanding of our contribution. Here are the new experiments:
> >
> > **Compare with other encoder methods.**
> > - In ALFworld, REVEAL-IT is trained and tested in the visual environment, which has no prior/extral knowledge from the text engine (text world in ALFworld). This is the same for the baseline methods (BLIP-2, LLaMA-Adapter, InstructBLIP and MiniGPT-4.)
> > - To compare with other encoder methods, we need to train REVEAL-IT in the text engine (without visual environment), and also we need to compare it with the baselines in the same conditions. Therefore, we have added new experiments for comparison with LLM-based baselines. We report the results in the table below, and we have also include it in the paper. Please note that the experiment setting is different with the conditions in Table.1.
> >
> > |Methods| Avg.|Pick|Clean|Heat|Cool|Look|Pick2|
> > |:--|:--|:--|:--|:--|:--|:--|:--|
> > |REVEAL-IT| 0.86|0.72|0.96|**0.87**|**0.82**|**0.95**|**0.90**|
> > |ReAct|0.54|0.71|0.65|0.62|0.44|0.28|0.35|
> > |AutoGen|0.77|0.92|0.74|0.78|0.86|0.83|0.41|
> > |Reflextion|**0.91**|**0.96**|**1.00**|0.79|**0.82**|0.94|0.88|
> >
> > **ReAct:** Shunyu Yao, Jeffrey Zhao, Dian Yu, Nan Du, Izhak Shafran, Karthik R. Narasimhan, and Yuan Cao. React: Synergizing reasoning and acting in language models. In ICLR, 2023.
> >
> > **Reflextion:** Noah Shinn, Federico Cassano, Ashwin Gopinath, Karthik RNarasimhan, and Shunyu Yao. Reflexion: Language agents with verbal reinforcement learning. In NeurIPS, 2023.
> >
> > **AutoGen:** Qingyun Wu, Gagan Bansal, Jieyu Zhang, Yiran Wu, Shaokun Zhang, Erkang Zhu, Beibin Li, Li Jiang, Xiaoyun Zhang, and Chi Wang. Autogen: Enabling next-gen llm applications via multi-agent conversation framework.
> >
> > **REVEAL-IT in Mujoco.**
> > - First, we want to emphasize that the targeting problem of REVEAL-IT is to provide explanations for the agent learning process in complex environments and tasks. The policy structure visualization can be easy for human to understand based on the task discription. We include the experiments in OpenAI GYM environment is to show that REVEAL-IT can also work/bring improvment to the basic RL algorithms in general environments.
> > - We have conducted new experiments per your request in mujoco. Due to the limited time, we can only compare with two baselines, and we will include the whole benchmark in the future. Here are the results:
> >
> > |Agent| Ant-v3 |Swimmer-v3|Hopper-v3|HalfCheetah-v3|
> > |:--|:--|:--|:--|:--|
> > |REVEAL-IT+PPO|2745.57 $\pm$ 564.23|340.58 $\pm$ 6.20|2167.90 $\pm$ 102.81|6047.82 $\pm$ 87.21|
> > |PPO|1480.47 $\pm$ 407.39|281.78 $\pm$ 11.86|2410.11 $\pm$ 9.86|5836.27 $\pm$ 171.68|
> > |A2C|-15.93 $\pm$ 6.74|199.91 $\pm$ 1.32|679.01 $\pm$ 302.76|3096.61 $\pm$ 82.49|
> >
> > Considering we are approaching the end of the discussion phase, we kindly ask for your confirmation on whether our responses align with your expectations or if there are additional clarifications we could provide. Your insights have been immensely helpful in improving our work, and we sincerely appreciate the time and effort you’ve dedicated to reviewing.
> >
> > Looking forward to hearing from you.
> >
> > Best regards,
> >
> > Authors

---

> ### Comment · Reviewer_jsXE · 2024-11-30
>
> Thank you for your responses.
>
> >Regarding to the GYM environment, we agree that there are no obvious sub-tasks in GYM or similar environments. Instead of using random sample from the replay buffer (PPO or other online RL methods), we store the trajectories that bring the agent higher learning progress and do the experience replay based on these trajectories
>
> Thanks for explaining; this explanation appears to be missing from your paper. I think it's important to include it.
>
> >We have already provided learning objectives for the GNN explainer in Appendix C in the original edition. And now, we move this part to Line.310-320.
>
> I find the description you provided in lines 310-320 to be unclear- it looks like you provide an equation for the traditional GNN explainer learning objective, but I do not see an equation for the objective used by REVEAL-IT
>
> You did not answer my question from my review about Figure 2:
> >How were the portions of the policy that are common to several sub-tasks identified?
>
> In addition, you did not address the main weaknesses I raised with this paper- the writing is still very difficult to follow, and I do not see how the your method provides any helpful human-interpretability to the deep RL models. Could the authors provide concrete examples of helpful, actionable insights they gained from looking at the outputs of the GNN Explainer?

---

> > ### Author Response · Authors · 2024-11-30
> > **Thank you for the response.**
> >
> > Thanks for your response. We are glad to hear that our reply has addressed some of your concerns. Now, we further address your concerns as follow:
> >
> > **The learning objective of GNN explainer in REVEAL-IT.**
> > > I find the description you provided in lines 310-320 to be unclear- it looks like you provide an equation for the traditional GNN explainer learning objective, but I do not see an equation for the objective used by REVEAL-IT.
> >
> > - First, we want to re-claim that the learning objective for the GNN explainer is the same as a traditional GNN explainer, since we formulate it as a GNN explanation problem. The difference is the dataset used for training GNN explainer is collected from the agent's learning process and evaluation.
> >
> > **Reviews about Figure 2.**
> > > How were the portions of the policy that are common to several sub-tasks identified?
> >
> > - Since we have a ReLU function to identify which part of the policy is activated during the evaluation, we can identify the shared part by comparing the high-lighted policy visualizations across different sub-tasks.
> > - To understand how the policy learns to complete a whole task, we design REVEAL-IT to visualize the policy update information for human to understand what a ability that the agent learns in a sub-task. To make this visualizion simpler and easy to read, we deploy a GNN explainer to highlight the important update. Based on this, by comparing the visualized results across different sub-tasks, human can understand which part of the policy maps to a specific ability to complete a necessary step for the whole task and which part of the policy corresponds to a sharing ability for diverse tasks.
> >
> > **REVEAL-IT help humans to understand agent's behaviour in complex environment.**
> >
> > > do not see how the your method provides any helpful human-interpretability to the deep RL models. Could the authors provide concrete examples of helpful, actionable insights they gained from looking at the outputs of the GNN Explainer?
> >
> > - The most direct way to understand a deep learning framework is to explain how the network works, like Grad-CAM to highlight the important part in a figure for recognition. However, this is not easy to achieve for RL especially in complex environments/tasks. This is because we cannot directly map a specific section of the policy to an agent's behavior. On the other hand, it is quite cost to compute the state value for every instance from the agent's view in a visual environment.
> > - We want to re-emphasize the limitations of traditional XRL methods in a complex environment, i.e., conterfacutal-based/causal-based methods **cannot work** in such a complex environment (e.g., ALFworld). More specifically, saliency mapping, causal RL and counterfacutal methods **require to intervening on the RL environment to generate counterfactual states/intervening data for conterfacual learning**. In Atari games, this would be fine to work since the environment is quite simple. But intervening a complex and continuous environment can be challenging. To handle this gap, we have tried saliency mapping (the source code provided in https://arxiv.org/pdf/1912.05743) in the past several days per reviewer mUUC's request. To train a conterfactual states generator, we create a dataset collected from 134 AlfWorld environments across six different tasks. Note that we only collect 8 pixel figures for each task in each environment, and the dataset reaches 4.4 GB. We have tried several ways but the generator cannot work well.
> > - Similar to saliency map, We visualize a value map based on the trained policy by REVEAL-IT in AFLworld to show the ability that the agent learns. Please refer to https://anonymous.4open.science/r/temporary-log-E0F5/README.md to check. We will include this in the next version. We hope this reply can help you understand the targetting problem of REVEAL-IT.

---

> ### Comment · Reviewer_jsXE · 2024-12-02
>
> Thanks for your response and further clarifications.
>
> > We visualize a value map based on the trained policy by REVEAL-IT in AFLworld to show the ability that the agent learns.
>
> How exactly do you compute this value map? And what insights does this provide?

---

> > ### Author Response · Authors · 2024-12-02
> > **Thank you for the response**
> >
> > Thank you for the response, and we are glad that our new response has addressed your concerns.
> >
> > To visualize a value map for continuous states in RL, we roll out the trajectories of the agents during the training, and we count the actions taken by the agent. After that, we highlight the agent-interact states to check how the agent tries to complete the task. The highlighted part indicates a higher frequency for the agent to interact with the environment. By comparing the highlighted part in the early stage of the training, we can tell the agent is learning to recognize the targeting item in the environment. However, we cannot understand the process of the agent learning to complete tasks solely based on these. It is just a statistical result, not a quantitative analysis. **We want to emphasize that the inherent limitation of the traditional XRL methods CANNOT work in complex environments, which we have always stressed, and deploying quantitative analysis in such environments and tasks remains an open problem.**
> >
> > In the saliency map, the environments are simple, and it is easy to calculate the state-value directly (e.g., we can directly use $V(s)$ for training, and we print $V(s)$ to explain the agent's behavior). However, policy-based methods are more common to use in complex environments/tasks, and visualizing the correponding state-value map is challenging. Meanwhile, this problem can be more complicated when the environment is more complex (e.g., mujoco is a continuous environment but much simpler than Alfworld).
> >
> > We hope this helps to address your concerns, and we are pleased to address any further concerns you may have. Thank you agian for the time and contribution to ICLR.

---

> > > ### Comment · Reviewer_jsXE · 2024-12-02
> > >
> > > Thanks for explaining- so the authors demonstrate that alternative visualization methods are not helpful. However, the reviewer still does not see how REVEAL-IT's visualization is helpful. It seems that the authors have yet to provide any examples of useful/actionable insights that can be gleaned from the outputs of the GNN explainer. The reviewer is also still concerned about the limits of this method for modern deep RL networks with large numbers of weights, where highlighting even a small fraction would still leave an unmanageable number of weights for a human to look at.

---

> ### Author Response · Authors · 2024-12-03
> **We disagree with your point.**
>
> Thanks for the response. We are glad to hear that we have reached a consensus on the limitations of traditional XRL methods. However, we disagree with your point that the GNN explainer is not helpful for understanding the agent's behavior when the policy  is large. Graph tasks widely adopt GNN pruning as a technique for explaining GNN, particularly in scenarios where the GNN network is too large. Pruning does not affect performance, and it can help humans understand the key nodes in the GNN. This is consistent with the purpose of REVEAL-IT. Through pruning, we can also understand the important nodes and corresponding weights in the policy, pay attention to their changes during training, and better understand the learning process of the agent. On the other hand, based on the understanding of the policy, REVEAL-IT also optimizes the task sequence for training, which achieves better performance and learning efficiency of RL.

---

### Official Review · Reviewer_mUUc · 2024-11-04

**Soundness:** 2
**Presentation:** 1
**Contribution:** 1
**Rating:** 3
**Confidence:** 4

**Summary:**

The paper presents an interpretability framework to understand an agent’s learning process in complex tasks (e.g., ALFWorld) through a GNN-based explainer. This method examines policy updates across predefined subtasks and highlights critical sections of the policy network.

**Strengths:**

the framework provides a structured approach for interpreting the learning progress of agents in long-horizon tasks using a GNN-based model.

**Weaknesses:**

- the paper's clarity could be improved. Certain terms are referenced repeatedly early in the text (e.g., introduction) but are defined too late or not at all—examples include "node-link diagram," "policy structure," "structure training tasks," and "problem" (line 91).
- the three claimed benefits of "understanding the agent’s performance post-training from the learning process of the policy and the sequences of training tasks" are difficult to grasp. Benefit 1 is too abstract, and lacks contextual detail. In contrast, Benefit 3 includes broad statements unsupported by references (e.g., "which can not deal with the big and complex problems that can not be seen in the real world").
- several key concepts lack references, including SCM and counterfactual methods (lines 95-96), MDP (line 167), and the node-link diagram representation (line 162).
- the paper motivates the question "why an agent can succeed or fail in a task" but lacks examples or case studies that would provide a unique takeaway on RL agents' interpretability.
- section 3’s "Structural visualization of the policy" is hard to understand. Goals are listed, but it is unclear how they are grounded or justified. For instance, it is mentioned that the policy visualization should use a node-link diagram to depict network architecture, but the rationale behind this choice is not explained. Additionally, it is unclear how this visualization allows users to judge the network’s robustness to translational and rotational variances or ambiguous inputs. The "gap" between visualization requirements and actual results remains unaddressed.
- in Figure 1, the authors introduce the GNN-explainer as part of the proposed framework, but Section 4.2 later introduces a GNN-predictor (also in Algorithm 1) without clarifying where it fits within Figure 1, creating confusion.

- the related work in explainable reinforcement learning (XRL) is not up-to-date, lacking recent advances in XRL.
- given that this work offers neuron-level visualization, it would benefit from referencing related literature in mechanistic interpretability (which is for understanding the inner workings of neural networks).
- the claim that prior explanation algorithms cannot model complex behaviours (lines 44-47) lacks evidence. Although (Puiutta & Veith, 2020) is cited to support this claim, it is a survey paper, which weakens the argument.

- how do the authors ensure there is any semantical interpretation w.r.t. part of the policy weights (so that humans can understand) when using GNN-explainer to visualise the policy (section 4.2)? in other words, how could users understand the visualised section of the policy? how could users link the "part of the edges (updated weights)" to the success of the RL agent?

- the GNN-based explainer is suggested to provide an understanding of each subtask’s value in training, yet this explanation seems limited to high-level progress indicators rather than deep rationales behind actions. This contradicts some of the authors’ statements like "a proficient explanation enhances understanding of the agent’s actions and helps improve performance" (lines 60-62). Moreover, the reliance on predefined subtasks limits the framework's applicability in real-world scenarios.

- step 1 in Section 4.2 is difficult to follow, particularly the authors' claim that variability does not affect GNN training. Additionally, the connection between "nodes linked to significant updates" and "activated nodes during the test" remains unclear. The assertion that "REVEAL-IT is distinguished from other explainable RL methods by its independence from external data and structure" is also debatable, as saliency maps do not impose environment or algorithm-specific constraints either.

- in Algorithm 1, it is unclear how the GNN optimizes the training task sequence; the sequence sampling appears to be based only on $P$ (see line 7 in Algorithm 1).

- a brief comparison of REVEAL-IT with baselines is missing, which is important for understanding the reasons behind its performance advantages—whether due to improved planning steps or better-learned low-level behaviours.

- figure 4, relevant to the discussion in Section 5.2, is placed in the appendix. Moving it (or parts of it) to the main text would improve readability and flow.

- the first question in Section 5 ("learning process of an RL agent") does not appear to be fully answered. It’s unclear where this process is visualized—Figure 2 or Figure 3. How could the nodes in Figure 2 be interpretable for users, what are the verbs in Figure 3 (are they subtasks?) and which final task is Figure 3 about?

**Questions:**

- in Section 5, what is the precise format of the explanation the authors intend to provide? Is the optimal training task sequence itself considered the explanation, as suggested in Section 5.2 (lines 429-471)?

- what is the objective of the controller, and what purpose does the control policy serve? This remains unexplained.

- in lines 100-102, does "updating the policy" equate to "updating the agent’s learning process"? Could the authors clarify this distinction?

- could the authors elaborate on the terms “nodes linked to significant updates” and “activated nodes during the test” in Section 4.2, specifically how their correlation is analyzed?

- where is Figure 3 referenced in the main text?

---

> ### Author Response · Authors · 2024-11-22
> **Thank you for the review! We hope our reply can address your concerns!**
>
> Thank you for your time and valueable suggestions! We have updated the paper following your suggestions, please refer to the highlighted part in the rebuttal revision. Now, We address your main concerns as below:
>
> **The basic definitions in RL.**
> > Question 2: what is the objective of the controller, and what purpose does the control policy serve?
> > Question 3: does "updating the policy" equate to "updating the agent’s learning process"?
> - The controller is the control policy $\pi_t$, which is used to control the RL agent to complete tasks in the environment.
> - The learning objective of $\pi_t$ is following the standard setting in RL, i.e., we train it to maxmize the accumalted reward $G_T$, which is defined as: $G=\sum_{t=0}^{T} \gamma^t R_{t+1}$, where $T$ denotes the maximal timesteps for each trajectory.
> - Yes. Updating the policy is equal to the learning process of the RL agent. We will change the expression here. Thank you for the suggestion!
>
> **The explanations provided by REVEAL-IT based on the activated nodes.**
> > Question 1: what is the precise format of the explanation the authors intend to provide? Is the optimal training task sequence itself considered the explanation, as suggested in Section 5.2 (lines 429-471)?
> > Question 4: could the authors elaborate on the terms “nodes linked to significant updates” and “activated nodes during the test” in Section 4.2, specifically how their correlation is analyzed?
> - REVEAL-IT is able to provide explanations in complex environments and tasks differs that the other conventional explanation methods or causal RL methods CANNOT (they usually target simple 2D environments, such like GYM, roboschool).
> - In the tesh phase, some nodes in the control policy will be actviated to control the agent to complete the task. These nodes are strongly related to the specific capabilities of the agent. REVEAL-IT explains in which sub-tasks the weights of the links to these nodes are trained and learned, so the update of these weights is "significant updates". Combined with the sub-task sequence, we can understand how the agent gradually learns to complete a complex task in a series of sub-tasks.
> - Therefore, we can claim that REVEAL-IT explains the learning process of an RL agent based on the control policy, i.e., REVEAL-IT learns to highlight the important update in the weights of the control policy during the training. In Fig.2, the GNN explainer in REVEAL-IT learns to highlight which parts of the updated weights is important to the success of completing each sub-task (note the thicker gray lines in each graph in fig.2). By comparing the shared important udpates across different sub-tasks, we can understand if there is some shared abilities for the agent in different tasks (note the organe box in fig.2).
>
> **Reference effor**
> > where is Figure 3 referenced in the main text?
> - We discuss the Fig.3 in the paragraph starts from Line 429. We apologize for the latex reference error, and we have updated this part.

---

> > ### Author Response · Authors · 2024-11-22
> > **Reply to weakness.**
> >
> > **Weakness**
> > 1. `several key concepts lack references`
> > - Thank you for pointing this out. We understand the missed references might cause some misunderstanding or confuses to the readers who are not familiar with this field. We have added the references following your suggestions.
> >
> > 2. `The limitations of counterfactual or causal methods`
> > - The main reason that the traditional conterfacutal/causal methods can't work in complex environments/tasks is: constructing a true causal model/SCM relies on a perfect understaning of the data generation process, any correlation, intervention and even counterfactual inquiries. However, given the inherent complexity of the world, it is often impractical to access a fully specified SCM. This claim has been fully discussed in previous works and survey papers[1], [2].
> > [1] Bernhard Schölkopf, Francesco Locatello, Stefan Bauer, Nan Rosemary Ke, Nal Kalchbrenner, Anirudh Goyal, and Yoshua Bengio. Toward causal representation learning.
> > [2] Jean Kaddour, Aengus Lynch, Qi Liu, Matt J. Kusner, and Ricardo Silva. Causal machine learning: A survey and open problems.
> >
> > 3. `Figure 1 creats some confusion`
> > - Thank you for the reminding. We have updated this figure following your suggestion.
> >
> > 4. `It is unclear why we need policy visualization in REVEAL-IT and How to understand the explanations`
> > - As discussed above, it is challenging to understand what a specific ability that the agent learns in a complex world/task through causal RL/counterfactural methods, which means an incomplete or wrong SCM will lead to bias.
> > - Based on this premise, we explored whether there is an explanation mechanism that can explain the agent's learning process **without introducing external bias**. We found that we could explain the agent's learning process through the update of the policy itself. By visualizing the update of the policy weights and the specific activation status when evaluating the policy, we can understand what capabilities the agent has learned.
> > - However, in complex tasks and environments, the structure of the policy itself may be very complex (multi-layer neural network and huge weight matrix), and there might be millions of policy updates for training. Even if we visualize all updates, this explanation is undoubtedly **unreadable** to humans. This is why we designed a GNN explainer to simplify and highlight "important updates" to assist humans in understanding.

---

> > > ### Author Response · Authors · 2024-11-25
> > > **Thank you for your effort to ICLR! Please check our rebuttal. We are looking forward to hear from you.**
> > >
> > > Thank you again for your valuable review of our submission. We have carefully considered your comments and incorporated responses to address the concerns raised. In particular, we clarify the unclear points and definitions in the paper and your concerns about the explanation methods. We hope our rebuttal provides sufficient detail to address your points effectively.
> > >
> > > We have also conducted new experiments per the reviewer tFxe and Lyow's requests. We hope the new experiments can help to strengthen your understanding of our contribution. Here are the new experiments:
> > >
> > > **Compare with other encoder methods.**
> > > - In ALFworld, REVEAL-IT is trained and tested in the visual environment, which has no prior/extral knowledge from the text engine (text world in ALFworld). This is the same for the baseline methods (BLIP-2, LLaMA-Adapter, InstructBLIP and MiniGPT-4.)
> > > - To compare with other encoder methods, we need to train REVEAL-IT in the text engine (without visual environment), and also we need to compare it with the baselines in the same conditions. Therefore, we have added new experiments for comparison with LLM-based baselines. We report the results in the table below, and we have also include it in the paper. Please note that the experiment setting is different with the conditions in Table.1.
> > >
> > > |Methods| Avg.|Pick|Clean|Heat|Cool|Look|Pick2|
> > > |:--|:--|:--|:--|:--|:--|:--|:--|
> > > |REVEAL-IT| 0.86|0.72|0.96|**0.87**|**0.82**|**0.95**|**0.90**|
> > > |ReAct|0.54|0.71|0.65|0.62|0.44|0.28|0.35|
> > > |AutoGen|0.77|0.92|0.74|0.78|0.86|0.83|0.41|
> > > |Reflextion|**0.91**|**0.96**|**1.00**|0.79|**0.82**|0.94|0.88|
> > >
> > > **ReAct:** Shunyu Yao, Jeffrey Zhao, Dian Yu, Nan Du, Izhak Shafran, Karthik R. Narasimhan, and Yuan Cao. React: Synergizing reasoning and acting in language models. In ICLR, 2023.
> > >
> > > **Reflextion:** Noah Shinn, Federico Cassano, Ashwin Gopinath, Karthik RNarasimhan, and Shunyu Yao. Reflexion: Language agents with verbal reinforcement learning. In NeurIPS, 2023.
> > >
> > > **AutoGen:** Qingyun Wu, Gagan Bansal, Jieyu Zhang, Yiran Wu, Shaokun Zhang, Erkang Zhu, Beibin Li, Li Jiang, Xiaoyun Zhang, and Chi Wang. Autogen: Enabling next-gen llm applications via multi-agent conversation framework.
> > >
> > > **REVEAL-IT in Mujoco.**
> > > - First, we want to emphasize that the targeting problem of REVEAL-IT is to provide explanations for the agent learning process in complex environments and tasks. The policy structure visualization can be easy for human to understand based on the task discription. We include the experiments in OpenAI GYM environment is to show that REVEAL-IT can also work/bring improvment to the basic RL algorithms in general environments.
> > > - We have conducted new experiments per your request in mujoco. Due to the limited time, we can only compare with two baselines, and we will include the whole benchmark in the future. Here are the results:
> > >
> > > |Agent| Ant-v3 |Swimmer-v3|Hopper-v3|HalfCheetah-v3|
> > > |:--|:--|:--|:--|:--|
> > > |REVEAL-IT+PPO|2745.57 $\pm$ 564.23|340.58 $\pm$ 6.20|2167.90 $\pm$ 102.81|6047.82 $\pm$ 87.21|
> > > |PPO|1480.47 $\pm$ 407.39|281.78 $\pm$ 11.86|2410.11 $\pm$ 9.86|5836.27 $\pm$ 171.68|
> > > |A2C|-15.93 $\pm$ 6.74|199.91 $\pm$ 1.32|679.01 $\pm$ 302.76|3096.61 $\pm$ 82.49|
> > >
> > > Considering we are approaching the end of the discussion phase, we kindly ask for your confirmation on whether our responses align with your expectations or if there are additional clarifications we could provide. Your insights have been immensely helpful in improving our work, and we sincerely appreciate the time and effort you’ve dedicated to reviewing.
> > >
> > > Looking forward to hearing from you.
> > >
> > > Best regards,
> > >
> > > Authors

---

> > > ### Comment · Reviewer_mUUc · 2024-11-25
> > >
> > > Thanks for the updates from the authors and I appreciated the extra experiments. Following are my responses:
> > >
> > > It seems there is a misunderstanding of my primary concerns. I want to clarify that my main concern is NOT about the basic definitions of reinforcement learning (RL)—the issue is the inconsistent usage of "controller" and "policy" in the original manuscript. Additionally, it is NOT about the reference to Figure 3. Instead, the issue lies with, to name a few, the lack of reference to key concepts when discussing other XRL techniques (e.g., SCM, counterfactual reasoning, and MDP), which remains missing; broad statements (e.g., Benefit 3) are still unsupported by references, and the claim that prior explanation algorithms cannot model complex behaviours lacks evidence (see more below).
> > >
> > > My **main concerns** (rephrased and copied from my original review):
> > >
> > > - The authors claim that their framework provides explanations in complex environments where other XRL techniques fail. This **claim is only partially true**. Specifically, the paper needs to clarify why other XRL techniques (e.g., saliency maps, reward decomposition, causal models) cannot handle the environment used in the experiments. Without testing these methods in the same domain, the exclusion of their applicability remains unsupported. Readers need evidence detailing why these methods are unsuitable and how the proposed framework overcomes those limitations. Simply stating that these techniques cannot be tested in 3D environments without tests is insufficient, especially when the claimed contribution is around explainability.
> > >
> > > - Another notable issue is **the lack of baselines for comparison with other XRL techniques**. Instead, the comparisons are made to non-XRL methods (GNNexplainer and MixupExplainer don't fall into this as not for RL), and the evaluation metric (success rate) is uncommon for XRL studies. This raises several questions: What advantages does the proposed interpretability framework provide over existing XRL techniques? What evaluation metrics are suitable for this comparative analysis? Why is there no qualitative analysis of interpretability, a critical aspect of XRL research?
> > >
> > > - The authors claim that their framework highlights important updates in the weights of the control policy during training. However, it is **unclear how users can semantically interpret the visualized sections of the policy weights**. Specifically: How can users relate the "highlighted edges" (updated weights) to the RL agent's success? How does this visualization enhance human understanding of the agent's behaviour? **The paper emphasizes task performance, but the highlighted updates (from GNN-explainer) seem far from human-interpretable**. Despite this, the authors position "update highlights" as a key contribution (Sections 1 and 3), which seems overstated given the lack of a user-centric interpretation framework
> > >
> > > - The paper motivates the question, "Why does an agent succeed or fail in a task?" but does not provide examples or case studies to address this. Instead, **the experiments solely focus on task performance, with no study of explainability**. If the focus is on training efficiency and performance, the paper should avoid repeatedly framing the contribution around explainability, especially without including qualitative analysis in the experiments
> > >
> > > Another **concern** (seemed neglected in the original review):
> > >
> > > - the claim that "REVEAL-IT is distinguished from other explainable RL methods by its independence from external data and structure" is debatable. For instance, saliency maps also operate without imposing environment or algorithm-specific constraints.
> > > Understanding performance advantages:
> > >
> > > - in Section 5.2, even when analyzing training efficiency and performance, it is unclear whether the advantages stem from improved planning steps or better-learned low-level behaviours, making it difficult to understand the reasons behind the observed performance gains, this applies to the new experiments as well.

---

> ### Author Response · Authors · 2024-11-30
> **Thank you for the response.**
>
> Thank you for the response. We address your concerns as below:
>
> **Compare with other RL techniques.**
> > The authors claim that their framework provides explanations in complex environments where other XRL techniques fail. This claim is only partially true. Specifically, the paper needs to clarify why other XRL techniques (e.g., saliency maps, reward decomposition, causal models)
> - The main reason we didn't include the comparison with other XRL techniques is that conterfacutal-based/causal-based methods **cannot work** in a complex environment (e.g., ALFworld). More specifically, saliency mapping, causal RL and counterfacutal methods **require to intervening on the RL environment to generate counterfactual states/intervening data for conterfacual learning**. In Atari games, this would be fine to work since the environment is quite simple. But intervening in a complex and continuous environment can be challenging.
> - More specifically, we want to argue that we are not targeting the same problem as the saliency map. It is intuitive to use state-value to reflect and analyze the agent's behavior in a discrete environment. However, this is not the same thing in a complex and continuous environment, i.e., we have tried saliency mapping (the source code provided in https://arxiv.org/pdf/1912.05743) in the past several days per your request. We create a dataset collected from 134 AlfWorld environments across six different tasks. Note that we only collect 8 pixel figures for each task in each environment, and the dataset reaches 4.4 GB. We tried to train a generator to generate counterfactual states based on it but failed.
> - We visualize a value map based on the trained policy by REVEAL-IT in AFLworld to show the ability that the agent learns. Please refer to https://anonymous.4open.science/r/temporary-log-E0F5/README.md to check. We will include this in the next version. We appreciate your suggestions, and we believe this would help in understanding the advantages of our method. We hope this reply can help you understand the target problem of REVEAL-IT.
>
> **Human can understand the agent's learning by comparing the visualized policy update across tasks.**
> > it is unclear how users can semantically interpret the visualized sections of the policy weights. Specifically: How can users relate the "highlighted edges" (updated weights) to the RL agent's success?
> - The most direct way to understand a deep learning framework is to explain how the network works, like Grad-CAM to highlight the important part in a figure for recognition. However, this is not easy to achieve for RL especially in complex environments/tasks. This is because we cannot directly map a specific section of the policy to an agent's behavior. On the other hand, it is quite cost to compute the state value for every instance from the agent's view in a visual environment.
> - To understand how the policy learns to complete a whole task, we designed REVEAL-IT to visualize the policy update information for humans to understand what an ability  the agent learns in a sub-task. To make this visualization simpler and easier to read, we deploy a GNN explainer to highlight the important update. Based on this, by comparing the visualized results across different sub-tasks, humans can understand which part of the policy maps to a specific ability to complete a necessary step for the whole task and which part of the policy corresponds to a sharing ability for diverse tasks.
>
> **Saliency map**
> > the claim that "REVEAL-IT is distinguished from other explainable RL methods by its independence from external data and structure" is debatable. For instance, saliency maps also operate without imposing environment or algorithm-specific constraints. Understanding performance advantages:
> - To our best knowledge, saliency methods are not designed to formalize an abstract human-understandable concept, and they do not provide a means to quantitatively compare semantically meaningful consequences of agent behavior. This leads to subjectivity in the conclusions drawn from saliency maps. Usually, in the RL community, the saliency map relies on the counterfacutal analysis to enhance its subjectivity.
>
>
> **REVEAL-IT does not do planning.**
> >it is unclear whether the advantages stem from improved planning steps or better-learned low-level behaviours
> - REVEAL-IT only optimizes the sub-task sequences for training rather than planning. We apologize if we misunderstand the meaning of "planning steps" here. If the "planning steps" means the sub-task planning, we think the two terms have the same meaning, since a better sub-task sequence will make the agent learn basic abilities more efficiently and results in a better performance and improved learning efficiency.

---

### Author Response · Authors · 2024-11-23
**Thank you for the time and suggestions. We are looking forward to hear from you.**

We sincerely thank all the reviewers for their valuable feedback and thoughtful suggestions, which have significantly helped us improve the clarity, depth, and presentation of our work. We have carefully addressed each of the concerns raised, incorporating revisions and additional explanations to ensure our responses are comprehensive and satisfactory. Now, to save your time, we summarize the main concerns and the response as below:

1. **Reviewer mUUC**:
- We have clarified the RL concepts and explanation format and terms in the new version.
- We have provided more detailed explanations on REVEAL-IT both in the rebuttal and the paper.
- We have provided explanations to your concerns in weakness.
2. **Reviewer JsXE**:
- We have clarified the definiton of "activated nodes" and the GNN explainer's learning objective.
- We explained how REVEAL-IT works in a environment without obvious sub-task structure.
- We explained that multi-modal challenges are beyond the current scope to keep the work focused but are potential future directions.
3. **Reviewer tFxe**:
- We have included new experiments based on different encoders per your request.
- We explained the motivations of using GNN explainer rather than alternative methods you mentioned.
- We explained the issues of alignment between the GNN explainer and RL policy.
4. **Reviewer Lyow**:
- We explained assumptions about sub-tasks and generality in environments without explicit sub-tasks.
- We clarified REVEAL-IT as an addition to existing algorithms, adaptable across RL methods.
- We acknowledged curriculum learning as a supporting tool rather than a key contribution and added a related work discussion to clarify this.

We hope that our updates and detailed responses effectively resolve the reviewers' concerns. However, if any issues remain or require further discussion, we are more than willing to engage in additional discussion to clarify and improve the work further. Thank you again for your time and effort in reviewing our paper.

---

### Meta-Review · Area_Chair_Ns3B · 2024-12-19

**Metareview:**

The reviewers generally agree that this paper, in its current state, is unfit for publication due to the unclear and, in many parts, incomplete presentation. I encourage the authors to clarify their writing. The authors motivate the method as an interpretability method, but ultimately do not provide strong evidence in their experimental sections on how the resulting GNN explainer can be used for interpretation nor the fidelity of this lens for interpretability. Instead, the method is evaluated primarily as a curriculum learning algorithm and compared against baselines which do not make use of comparable curricula. Overall, the core framing, writing quality, and experimental design should be improved before this work is ready for publication.

**Additional Comments On Reviewer Discussion:**

The reviewers were largely confused around the interpretability mechanisms behind the proposed method. Many reviewers point out this approach is more akin to a curriculum learning method than an interpretability method, and this confusion was not convincingly addressed during rebuttals.

---

### Decision · Program_Chairs · 2025-01-22

Reject